# AnyEdit: Edit Any Knowledge Encoded in Language Models

Houcheng Jiang[1]  Junfeng Fang[2][*]  Ningyu Zhang[3]  Mingyang Wan[4]
Guojun Ma[4]  Xiang Wang[1]  Xiangnan He[1][*]  Tat-Seng Chua[2]

## Abstract

Large language models (LLMs) often produce incorrect or outdated information, necessitating efficient and precise knowledge updates. Current model editing methods, however, struggle with long-form knowledge in diverse formats, such as poetry, code snippets, and mathematical derivations. These limitations arise from their reliance on editing a single token's hidden state, a limitation we term as "efficacy barrier". To solve this, we propose **AnyEdit**, a new autoregressive editing paradigm. It decomposes long-form knowledge into sequential chunks and iteratively edits the key token in each chunk, ensuring consistent and accurate outputs. Theoretically, we ground AnyEdit in the Chain Rule of Mutual Information, showing its ability to update any knowledge within LLMs. Empirically, it outperforms strong baselines by 21.5% on benchmarks including UnKEBench, AKEW, and our new **EditEverything** dataset for long-form diverse-formatted knowledge. Additionally, AnyEdit serves as a plug-and-play framework, enabling current editing methods to update knowledge with arbitrary length and format, significantly advancing the scope and practicality of LLM knowledge editing. Our code is available at: `https://github.com/jianghoucheng/AnyEdit`.

## 1. Introduction

Large language models (LLMs) have achieved impressive success by learning and storing vast amounts of knowledge (Brown et al., 2020; Radford et al., 2019; Zhao et al., 2024). However, they often suffer from hallucinations, producing

incorrect or outdated information (Cao et al., 2021; Mitchell et al., 2022a). For instance, when queried with "Where were the latest Olympics held?", an LLM often provides an outdated response "Tokyo", instead of the correct, updated answer "Paris". While retraining or fine-tuning can mitigate these issues, such approaches are resource-intensive and risk overfitting (Mitchell et al., 2022b; Meng et al., 2022). To overcome this, *model editing* has emerged as a promising alternative. As illustrated in Figure 1 (b), it typically begins by constructing a (subject, relation, object) triplet, such as (Olympics, were held in, Paris), to represent the knowledge to be updated. It then follows a locate-then-edit paradigm as Figure 1 (a) shows: (1) Locate the key token in the input prompt (*e.g.*, "Olympics") and the influential layers using causal tracing; (2) Edit the hidden states of the key token within these layers to align the model's output with the desired knowledge update (*e.g.*, modifying "Tokyo" to "Paris"). This approach enables precise and efficient updates without the need for full-scale retraining or fine-tuning, showing the potential to model dynamic and evolving knowledge.

Despite their success, existing model editing methods mostly face significant limitations in the length and diversity of the knowledge they can update. As shown in Figure 1 (c), even leading methods like AlphaEdit (Fang et al., 2025) and RECT (Gu et al., 2024) struggle to handle updates exceeding 100 tokens. Worse still, most methods are restricted to the knowledge represented as structured (subject, relation, object) triples. However, real-world knowledge is often encoded in diverse formats (*e.g.*, mathematical derivations and code snippets as shown in Figure 1 (f)) and could exceed the 100-token threshold (Wu et al., 2024). These constraints are ill-suited for real-world scenarios, significantly narrowing the scope of model editing and hindering its broader advancement.

Here we first conduct an in-depth analysis to identify why current methods fail for **long-form diverse-formatted knowledge**. Considering Figure 1 (a) again, existing methods typically rely on locating a single token, assuming that altering its hidden states will suffice to edit the LLM's output (*i.e.*, enabling the model to generate desired outputs reflecting new knowledge). However, long-form diverse-formatted knowledge is inherently more complex and information-dense than a single triplet, often requiring the integration

*Corresponding Author. [1]MoE Key Lab of BIPC, University of Science and Technology of China [2]National University of Singapore [3]Zhejiang University [4]Douyin Co., Ltd. Correspondence to: Junfeng Fang <fangjf1997@gmail.com>, Xiangnan He < xiangnanhe@gmail.com>.

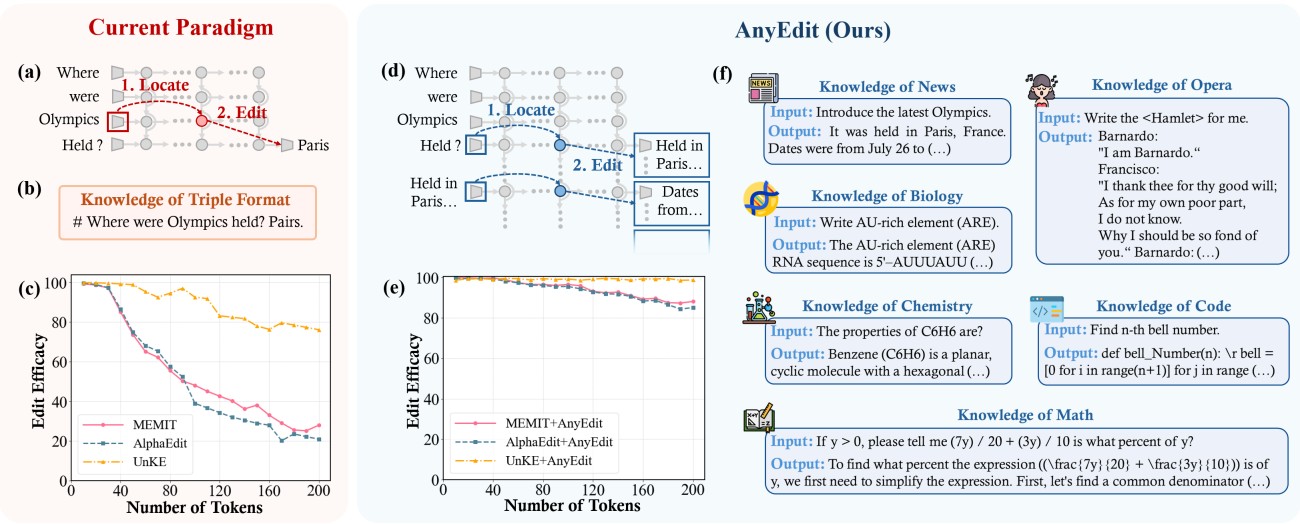

Figure 1. Comparison of current methods and our AnyEdit. (a) and (d) illustrate the editing processes; (c) and (e) show the editing efficacy as the number of tokens within the to-be-updated knowledge increases; (b) and (f) depict the type of knowledge that each method can edit.

of multiple critical tokens and intricate interdependencies among their hidden states. Thus, altering just a single token's hidden state is insufficient to ensure consistent and accurate knowledge generation. We term this limitation as the *efficacy barrier* of single-token editing[1], which is empirically validated in Section 3. As a consequence, existing methods remain constrained by the paradigm of locating a single token and, as a result, are unable to overcome the aforementioned limitation.

Hence, a critical question naturally arises for updating long-form diverse-formatted knowledge: "*Can multiple tokens be jointly located and edited to enable complex knowledge updates?*" A straightforward solution is to directly extend single-token editing to multiple tokens. However, it risks interference or conflicts between hidden state perturbations, undermining the coherence of to-be-updated knowledge and causing the performance drop as Figure 1 (c) showcases. In sight of this, we introduce **AnyEdit**, an autoregressive editing paradigm that enables collaborative multi-token updates. As illustrated in Figure 1 (d), AnyEdit decomposes long-form knowledge into sequential chunks, treating each as independent sub-knowledge. During editing, we iteratively (1) locate the final token of the current chunk and (2) perturb its hidden states to maximize the likelihood of generating the subsequent chunk. Building on the Chain Rule of mutual information (Dobrushin, 1963) in Information Theory (Kullback, 1997), we theoretically demonstrate that this autoregressive process ensures the generation of consistent, complete long-form knowledge. AnyEdit offers two

key advantages: (1) Adaptivity: The number of edited tokens can be adaptively adjusted with the knowledge length, avoiding redundant edits. (2) Generality: It supports diverse knowledge formats (*e.g.*, poetry, code, math) by decoupling structure-specific constraints. By shifting from single-token to multi-token collaborative editing, AnyEdit overcomes the challenge of efficacy barrier, demonstrating the potential for practical, updating complex knowledge of diverse formats.

To validate AnyEdit, we conducted a comprehensive evaluation comparing it with leading model editing methods (*e.g.*, MEMIT (Meng et al., 2023), AlphaEdit (Fang et al., 2025), and UnKE (Deng et al., 2025)) on the prevailing LLMs such as Llama3-8B-Instruct[2] and Qwen2.5-7B-Chat (Yang et al., 2024). Beyond standard benchmark datasets (*e.g.*, Counterfact (Meng et al., 2022) and ZsRE (Meng et al., 2023)) that represent knowledge as triples, we curate **EditEverything**, a new benchmark for long-form knowledge in diverse formats. As shown in Figure 1 (f), this dataset includes entries up to 458 tokens — over twice the length of the longest sequences in existing benchmarks (*e.g.*, 156 tokens in AKEW (Wu et al., 2024)) — and spans multiple domains, including mathematics, news, code, and biochemistry. Results on EditEverything and standard benchmarks demonstrate that AnyEdit surpasses all baselines, achieving a 21.5% average improvement in editing accuracy with comparable computational overhead. Furthermore, as the first autoregressive editing framework, AnyEdit also enables seamless integration with existing locate-then-edit methods, as shown in Figure 1 (e). This plug-and-play capability equips traditional approaches with the ability to handle knowledge with arbitrary length and format, significantly broadening

---

[1]The "Efficacy" is the metric proposed by the model editing method ROME (Meng et al., 2022), aiming to evaluate the success rate of editing. For more details, please refer to Appendix A.2.

[2]https://llama.meta.com/llama3

the scope and practicality of LLM knowledge editing.

## 2. Preliminary

**Autoregressive LLMs.** LLMs learn and store knowledge through autoregressive token prediction. Formally, let $f$ denote a decoder-only LLM with $L$ layers processing the input sequence $X = (x_0, x_1, \cdots, x_T)$. At layer $l$, the hidden state for token $x_t$ is computed via the forward propagation:

$$
\begin{aligned}
\boldsymbol{h}_t &= \boldsymbol{h}_t^{-1} + \boldsymbol{a}_t + \boldsymbol{m}_t, \\
\boldsymbol{a}_t &= \text{Attention}(\boldsymbol{h}_0^{-1}, \boldsymbol{h}_1^{-1}, \ldots, \boldsymbol{h}_t^{-1}), \\
\boldsymbol{m}_t &= \text{MLP}(\boldsymbol{h}_t^{-1} + \boldsymbol{a}_t),
\end{aligned}
\tag{1}
$$

where $\boldsymbol{h}_t$ and $\boldsymbol{h}_t^{-1}$ denote the hidden states of token $x_t$ in the current and previous layers, respectively; $\boldsymbol{a}_t$ and $\boldsymbol{m}_t$ are the outputs of the attention and MLP modules, respectively.

**Model Editing in LLMs.** To update outdated or incorrect knowledge within $f$, model editing typically follows a locate-then-edit paradigm (Meng et al., 2022): (1) Locate the key token in the input prompt and the influential layers; (2) Edit the hidden states of the key token within these layers to modify the model's output. Formally, let $(X, Y)$ denote the to-be-updated knowledge with the input prompt $X$ (*e.g.*, "Where were the latest Olympics held?") and the desired output $Y$ (*e.g.*, "Paris."). Suppose the key token is located at position $t$ in $X$. Current methods typically perturb its hidden state $\boldsymbol{h}_t$ by adding a residual term $\boldsymbol{\delta}$, which is obtained via gradient descent to maximize the probability of generating $Y$ given $X$:

$$
\boldsymbol{\delta} = \arg\min_{\hat{\boldsymbol{\delta}}} \left( -\log \mathbb{P}_{f(\boldsymbol{h}_t + \hat{\boldsymbol{\delta}})} \left[ Y \mid X \right] \right),
\tag{2}
$$

where $\mathbb{P}_{f(\boldsymbol{h}_t + \hat{\boldsymbol{\delta}})}$ represents the output probability when replacing $\boldsymbol{h}_t$ with $\boldsymbol{h}_t + \hat{\boldsymbol{\delta}}$ in the LLM. Finally, the LLM parameters are updated such that, given the input $X$, the hidden state of the key token is aligned with $\boldsymbol{h}_t + \boldsymbol{\delta}$. For more details, please refer to Appendix B.1.

## 3. Limitations of Single-token Editing

Although **single-token editing** methods have been extensively studied in recent years (Fang et al., 2025; Wang et al., 2024; Deng et al., 2025; Gu et al., 2024; Wu et al., 2024), we argue that they suffer from updating long-form, diverse formatted knowledge.

### 3.1. Editing Diverse-formatted Knowledge

According to Equation 2, the success of the single-token editing methods hinges on whether the LLM generates $Y$ given $X$ after applying the perturbation $\boldsymbol{\delta}$ to $\boldsymbol{h}_t$. In other words, the perturbation must significantly increase the probability

of generating $Y$ over any other possible output. However, if the original probability of $Y$ in the unedited LLM is already low — particularly common in the case of diverse-formatted knowledge, such as code snippets and mathematical derivations — then $\boldsymbol{\delta}$ must induce a large shift to make $Y$ the dominant output. Due to the limited capacity of single-token editing, current methods often struggle with such cases.

This limitation arises because structured knowledge (*e.g.*, factual triples) is simpler than diverse-formatted knowledge, which involves intricate dependencies. In triplets, modifying a single token (*e.g.*, changing "Tokyo" to "Paris") is often enough. In contrast, diverse-formatted knowledge like code and math requires consistent updates across multiple tokens due to syntax, variable dependencies, and hierarchical structures. Consider Figure 1 (f) again. Modifying a function in code may require changes across multiple lines, while altering a symbol in a formula often affects the entire expression. Single-token perturbations fail to capture and propagate such dependencies, leading to ineffective edits.

To empirically validate this, we apply leading single-token editing methods, MEMIT to update triplet-structured and diverse-formatted knowledge in LLaMA3-8B-Instruct. Figure 2 shows relationships between knowledge format, original probability, and editing efficacy. The results show that diverse-formatted knowledge, which typically has a low original probability, exhibits poor editing efficacy. In a nutshell, **low original probability** may be the fundamental limitation in updating **diverse-formatted** knowledge.

### 3.2. Editing Long-form Knowledge

Recent studies suggest that although LLMs leverage attention mechanism (Wang & Komatsuzaki, 2021), the dependency between distant tokens weakens as their positional distance increases (Liu et al., 2024a). As a result, for long outputs $Y$ (*e.g.*, $Y$ exceeding 100 tokens), perturbations applied to input tokens may have a diminishing influence on the later tokens within $Y$. In such cases, the shift of generation probability of $Y$ introduced by the perturbation $\boldsymbol{\delta}$ may be too small, making it insufficient to increase the probability of $Y$ above that of any other potential output.

To validate this, we repeat the experiments described in Section 3.1. Additionally, we use causal tracing (Meng et al., 2022), a common strategy in model editing, to quantify the shift in the generation probability of $Y$ introduced by the perturbation on $X$ (details of causal tracing can be found in Appendix B.1). Figure 3 illustrates the relationship between: the number of tokens in $Y$, the probability shift, and editing efficacy. The results demonstrate that long-form knowledge, which is less affected by single-token edits on the input, shows poor editing efficacy. In other words, the **low probability shift** introduced by single-token edits emerges as the inherent limitation in updating **long-form** knowledge.

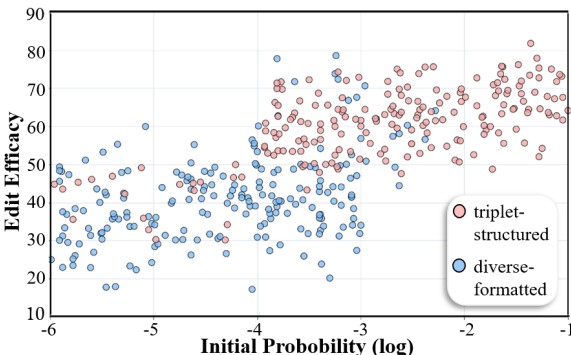

*Figure 2.* Relationship between knowledge format, original output probability, and efficacy when applying advanced editing methods to update triplet-structured and diverse-formatted knowledge. For each category, we randomly sample 200 knowledge instances to conduct experiments.

This limitation, in conjunction with the constraint discussed in Section 3.1, collectively suggests that the current single-token editing paradigm faces a theoretical efficacy barrier. We formalize this barrier as follows:

**Proposition 1** *(Efficacy Barrier of Single-token Editing). Given $N$ pieces of knowledge to be updated, denoted as $(X_i, Y_i)$ for $i \in (1, N)$, and a target LLM $f$, the upper bound on the efficacy of knowledge updates using single-token editing is given by:*

$$\eta = \frac{1}{N} \sum_{i=1}^{N} \operatorname{sign} \Big( \mathbb{P}_{f(\boldsymbol{h}+\boldsymbol{\delta})} \left[ Y_i \mid X_i \right] - \\ \max_{Y' \in \mathbb{Y}/Y_i} \left( \mathbb{P}_{f(\boldsymbol{h}+\boldsymbol{\delta})} \left[ Y' \mid X_i \right] \right) \Big), \tag{3}$$

*where $\operatorname{sign}(\cdot)$ is the sign function, $\boldsymbol{h}$ denotes the hidden state of the perturbed token, $\mathbb{Y}$ represents the set of $f$'s outputs, and $\boldsymbol{\delta}$ is defined as:*

$$\boldsymbol{\delta} = \arg\min_{\hat{\boldsymbol{\delta}}} \Big( - \log \mathbb{P}_{f(\boldsymbol{h}+\hat{\boldsymbol{\delta}})} \left[ Y_i \mid X_i \right] \Big). \tag{4}$$

*Here $\eta$ computes the average success rate of making $Y_i$ the most probable output for each input $X_i$ after editing.*

## 4. AnyEdit: Autoregressive Model Editing

The previous section demonstrated that, regardless of how current single-token editing methods are optimized, their editing efficiency is inevitably constrained by an upper bound. Furthermore, as the format and length of the knowledge to be updated become more diverse and longer, this upper bound diminishes, eventually rendering the editing process ineffective. To address this issue, we introduce AnyEdit, an autoregressive editing paradigm that enables collaborative token updates. The theoretical foundation from an information-theoretic perspective is provided in Section 4.1, while its instantiation is outlined in Section 4.2.

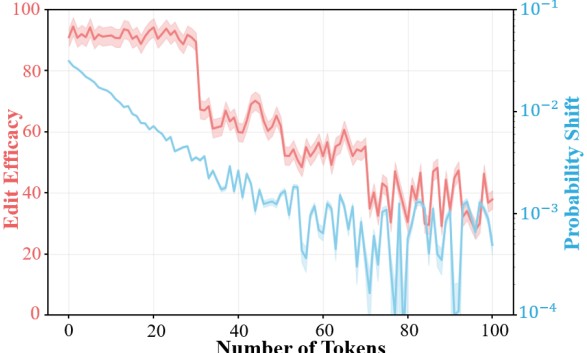

*Figure 3.* Relationship between the number of tokens in to-be-updated knowledge, probability shift under random perturbations, and efficacy. We conduct experiments by truncating the sampled knowledge instances to enable editing across different token lengths. The lighter-colored bands represent variance.

### 4.1. Theoretical Foundation

We begin by reviewing the editing process from an information-theoretic perspective. Specifically, existing methods aim to maximize the probability of generating $Y$ by editing hidden states, as formulated in Equation 2. This objective aligns with maximizing the mutual information (MI) (Kullback, 1997) between $X$ and $Y$, given the perturbed hidden state $\boldsymbol{h}'$:

$$\boldsymbol{h}' = \arg\max_{\hat{\boldsymbol{h}}'} I(X; Y \mid \hat{\boldsymbol{h}}'), \tag{5}$$

where $I(\cdot|\cdot)$ denotes MI. A detailed derivation is provided in Appendix B.2. Extending this framework to perturb the hidden states of $K$ tokens simultaneously, denoted as $\{\boldsymbol{h}_1, \boldsymbol{h}_2, \cdots, \boldsymbol{h}_K\}$, the MI in Equation 5 generalizes to:

$$I(X; Y \mid \boldsymbol{h}_1', \boldsymbol{h}_2', \cdots, \boldsymbol{h}_K'). \tag{6}$$

However, as discussed in Section 3.2, simultaneous perturbation of multiple hidden states introduces interference, reducing editing efficacy. Ideally, the MI term should involve only a single hidden state as a condition, as in Equation 5. To achieve this, we first decompose $Y$ into $K$ chunks, denoted as $Y = (Y_1, Y_2, ..., Y_K)$, and apply the chain rule of MI to rewrite Equation 6:

$$I(X; Y \mid \boldsymbol{h}_1', \cdots, \boldsymbol{h}_K') = \\ \sum_{k=1}^{K} I(X; Y_k \mid Y_1, \cdots, Y_{k-1}, \boldsymbol{h}_1', \cdots, \boldsymbol{h}_K'). \tag{7}$$

We then select the last token of each chunk as the perturbation target, establishing a one-to-one correspondence between chunk $Y_k$ and hidden state $\boldsymbol{h}_k'$. Given that LLMs generate outputs autoregressively, two key properties hold:

- Hidden states of later tokens do not influence the generation of earlier outputs, *i.e.*, $H(Y_p \mid \boldsymbol{h}_q') = H(Y_p)$ for $p < q$, where $H(\cdot)$ represents the information entropy;

- Once $Y_k$ is determined, conditioning on $Y_k$ subsumes conditioning on the hidden state of tokens within $Y_k$, *i.e.*, $H(\cdot \mid Y_k) = H(\cdot \mid Y_k, \boldsymbol{h}'_k)$.

Using these two properties, we can derive that each term in Equation 7 is proportional to:

$$I(X, Y_1, \ldots, Y_{k-1}; Y_k \mid \boldsymbol{h}'_k). \tag{8}$$

A detailed derivation is provided in Appendix B.3. Equation 8 directly informs our editing strategy: at each step, we take $X$ and previously edited chunks $(Y_1, \ldots, Y_{k-1})$ as inputs, iteratively encoding the next chunk $Y_k$ into the LLM's output in an autoregressive manner. More importantly, Equation 8 exhibits two critical advantages:

1. **Elimination of Interference:** Each MI term conditions on a single hidden state $\boldsymbol{h}'$, avoiding the interference issue in Equation 6.

2. **Scalability Across Length and Formats:** Regardless of $Y$'s length or format, each editing step updates only a single chunk, circumventing the efficacy barrier of single-token editing in Proposition 1.

In summary, this formulation provides a theoretical foundation for editing long-form diverse-formatted knowledge in LLMs. By leveraging an autoregressive editing paradigm, we move beyond single-token editing, enabling more scalable and efficient model editing. Next, we demonstrate how to instantiate it by providing specific implementation steps.

### 4.2. Implementation Details

Building on the above theoretical foundation, we elaborate on the implementation process of AnyEdit in this part. Specifically, AnyEdit follows a four-step process for effective and scalable knowledge editing:

**Step 1: Chunk Outputs.** The first step involves splitting the target output $Y$ into multiple chunks. We propose two chunking strategies: (1) a sliding window with a fixed number of tokens and (2) semantic segmentation based on natural sentence boundaries. These strategies endow AnyEdit with the ability to automatically adjust the number of edited tokens based on knowledge length, ensuring efficient edits without redundancy.

**Step 2: Locate Token and Layer.** We select the last token of each chunk $Y_k$ as the target for editing, and directly apply the causal tracing to locate the influential layers, following the conventional model editing methods (Meng et al., 2022).

**Step 3: Edit Hidden States.** Following Equation 8, we input $X$ and the previous chunks $\{Y_1, Y_2, \cdots, Y_{k-1}\}$ into the LLM. Then, the hidden state $\boldsymbol{h}_k$ of the selected token is

edited by the gradient descent to maximize the probability of generating $Y_k$. Formally,

$$\begin{aligned} \boldsymbol{h}'_k = &\boldsymbol{h}_k + \arg\min_{\hat{\boldsymbol{\delta}}} \\ &\left( -\log \mathbb{P}_{f(\boldsymbol{h}_k + \hat{\boldsymbol{\delta}})} \left[ Y_k \mid X, Y_1, \cdots, Y_{k-1} \right] \right). \end{aligned} \tag{9}$$

**Step 4: Update Model Parameters.** Finally, the LLM parameters are updated to align the hidden state of the selected tokens with the edited states, using standard least-squares optimization as employed in current model editing methods (Meng et al., 2022). Detailed implementation of this step is provided in Appendix B.1.

This multi-token collaborative editing process enables AnyEdit to overcome the efficacy barrier of single-token editing. Furthermore, AnyEdit enables seamless integration with existing methods, equipping them with the ability to edit any knowledge within LLMs and broadening the scope and practicality of LLM knowledge editing.

## 5. Experiments

In this section, we conduct extensive experiments to address the following research questions:

- **RQ1:** How does AnyEdit perform compared to other baselines on tasks involving long-form knowledge?

- **RQ2:** How does AnyEdit compare to other baselines in handling diverse-formatted knowledge?

- **RQ3:** Can AnyEdit improve the performance of other locate-then-edit methods?

- **RQ4:** How does the token length of each chunk affect the performance of long-form knowledge editing in AnyEdit?

### 5.1. Experimental Setup

In this subsection, we summarize the LLMs, baseline methods, datasets, and evaluation metrics used in our experiments. Further details are provided in Appendix A.

**LLMs & Baseline Methods.** We conducted experiments using two widely adopted LLMs: Llama3-8B-Instruct and Qwen2.5-7B-Instruct. For comparison with our method, we evaluated against several model editing methods, including FT-L (Zhu et al., 2020), MEND (Mitchell et al., 2022a), ROME (Meng et al., 2022), MEMIT (Meng et al., 2023), AlphaEdit (Fang et al., 2025), and UnKE (Deng et al., 2025).

**Datasets and Evaluation Metrics.** To evaluate the performance of unstructured long-form knowledge editing, we employed existing benchmarks, including UnKEBench (Deng et al., 2025) and AKEW (Wu et al., 2024). To further assess

*Table 1.* Long-form knowledge editing performance with different methods. "Pre-edited" refers to the unedited pre-trained LLM, and "Ori." and "Para." denote the outputs of the tested LLM for original questions and paraphrased questions respectively. The best results are highlighted in bold.

| LLM | Method | UnKEBench | | | | AKEW (Counterfact) | | | | AKEW (MQUAKE) | |
| --- | --- | --- | --- | --- | --- | --- | --- | --- | --- | --- | --- |
| | | Ori. | | Para. | | Ori. | | Para. | | Ori. | |
| | | Bert Score | Rouge-L | Bert Score | Rouge-L | Bert Score | Rouge-L | Bert Score | Rouge-L | Bert Score | Rouge-L |
| Llama3-8B-It | Pre-edited | $63.18_{\pm0.38}$ | $23.67_{\pm0.52}$ | $62.73_{\pm0.31}$ | $23.52_{\pm0.51}$ | $64.03_{\pm0.32}$ | $15.74_{\pm0.42}$ | $40.20_{\pm0.46}$ | $5.52_{\pm0.10}$ | $65.77_{\pm0.37}$ | $16.25_{\pm0.47}$ |
| | FT-L | $40.31_{\pm0.45}$ | $11.39_{\pm0.56}$ | $37.29_{\pm0.41}$ | $8.51_{\pm0.51}$ | $42.89_{\pm0.43}$ | $13.12_{\pm0.58}$ | $31.44_{\pm0.49}$ | $5.24_{\pm0.08}$ | $45.87_{\pm0.43}$ | $12.99_{\pm0.52}$ |
| | MEND | $68.73_{\pm0.34}$ | $29.24_{\pm0.52}$ | $64.11_{\pm0.32}$ | $28.05_{\pm0.54}$ | $68.81_{\pm0.31}$ | $30.30_{\pm0.48}$ | $41.56_{\pm0.40}$ | $10.95_{\pm0.57}$ | $67.85_{\pm0.33}$ | $22.48_{\pm0.56}$ |
| | ROME | $72.16_{\pm0.31}$ | $23.74_{\pm0.46}$ | $70.54_{\pm0.25}$ | $22.39_{\pm0.54}$ | $72.90_{\pm0.36}$ | $25.86_{\pm0.52}$ | $43.59_{\pm0.51}$ | $12.37_{\pm0.55}$ | $70.10_{\pm0.43}$ | $21.07_{\pm0.59}$ |
| | MEMIT | $76.21_{\pm0.36}$ | $30.49_{\pm0.52}$ | $74.25_{\pm0.31}$ | $28.65_{\pm0.61}$ | $76.44_{\pm0.33}$ | $32.20_{\pm0.48}$ | $47.80_{\pm0.34}$ | $16.09_{\pm0.59}$ | $75.31_{\pm0.37}$ | $22.73_{\pm0.61}$ |
| | AlphaEdit | $73.92_{\pm0.29}$ | $26.59_{\pm0.49}$ | $72.96_{\pm0.26}$ | $25.92_{\pm0.51}$ | $72.63_{\pm0.31}$ | $24.95_{\pm0.50}$ | $44.67_{\pm0.46}$ | $13.79_{\pm0.49}$ | $69.85_{\pm0.36}$ | $23.04_{\pm0.59}$ |
| | AnyEdit | $\mathbf{97.76_{\pm0.11}}$ | $\mathbf{92.96_{\pm0.24}}$ | $\mathbf{96.60_{\pm0.19}}$ | $\mathbf{95.60_{\pm0.35}}$ | $\mathbf{97.76_{\pm0.14}}$ | $\mathbf{95.87_{\pm0.23}}$ | $\mathbf{62.63_{\pm0.44}}$ | $\mathbf{46.51_{\pm0.59}}$ | $\mathbf{96.33_{\pm0.21}}$ | $\mathbf{94.32_{\pm0.23}}$ |
| | UnKE | $98.34_{\pm0.15}$ | $93.33_{\pm0.26}$ | $93.38_{\pm0.21}$ | $78.42_{\pm0.32}$ | $98.62_{\pm0.14}$ | $96.37_{\pm0.22}$ | $59.62_{\pm0.44}$ | $32.89_{\pm0.59}$ | $98.33_{\pm0.13}$ | $95.42_{\pm0.20}$ |
| | AnyEdit* | $\mathbf{99.86_{\pm0.08}}$ | $\mathbf{99.68_{\pm0.21}}$ | $\mathbf{94.70_{\pm0.12}}$ | $\mathbf{85.75_{\pm0.23}}$ | $\mathbf{99.95_{\pm0.01}}$ | $\mathbf{99.98_{\pm0.01}}$ | $\mathbf{64.24_{\pm0.48}}$ | $45.31_{\pm0.55}$ | $\mathbf{99.89_{\pm0.06}}$ | $\mathbf{99.69_{\pm0.09}}$ |
| Qwen2.5-7B-It | Pre-edited | $64.18_{\pm0.37}$ | $25.88_{\pm0.59}$ | $64.39_{\pm0.34}$ | $24.02_{\pm0.55}$ | $65.50_{\pm0.34}$ | $18.24_{\pm0.60}$ | $44.74_{\pm0.41}$ | $17.29_{\pm0.51}$ | $67.71_{\pm0.37}$ | $19.58_{\pm0.49}$ |
| | FT-L | $44.02_{\pm0.43}$ | $13.78_{\pm0.56}$ | $40.33_{\pm0.36}$ | $12.93_{\pm0.49}$ | $46.66_{\pm0.48}$ | $14.63_{\pm0.58}$ | $32.34_{\pm0.50}$ | $12.31_{\pm0.62}$ | $47.47_{\pm0.42}$ | $15.75_{\pm0.55}$ |
| | MEND | $69.49_{\pm0.38}$ | $27.77_{\pm0.61}$ | $62.01_{\pm0.44}$ | $27.92_{\pm0.57}$ | $69.54_{\pm0.54}$ | $25.47_{\pm0.49}$ | $52.86_{\pm0.40}$ | $22.81_{\pm0.54}$ | $69.40_{\pm0.32}$ | $32.39_{\pm0.44}$ |
| | ROME | $74.73_{\pm0.33}$ | $31.52_{\pm0.42}$ | $71.90_{\pm0.21}$ | $28.12_{\pm0.38}$ | $75.89_{\pm0.38}$ | $36.42_{\pm0.45}$ | $55.67_{\pm0.47}$ | $25.79_{\pm0.59}$ | $72.18_{\pm0.373}$ | $35.61_{\pm0.49}$ |
| | MEMIT | $78.03_{\pm0.30}$ | $38.04_{\pm0.47}$ | $76.50_{\pm0.31}$ | $28.65_{\pm0.50}$ | $77.19_{\pm0.32}$ | $38.95_{\pm0.48}$ | $56.04_{\pm0.40}$ | $25.73_{\pm0.57}$ | $73.15_{\pm0.32}$ | $34.39_{\pm0.54}$ |
| | AlphaEdit | $80.48_{\pm0.29}$ | $42.77_{\pm0.36}$ | $78.38_{\pm0.21}$ | $38.26_{\pm0.38}$ | $80.66_{\pm0.25}$ | $45.55_{\pm0.37}$ | $56.99_{\pm0.49}$ | $27.69_{\pm0.59}$ | $74.35_{\pm0.31}$ | $41.07_{\pm0.44}$ |
| | AnyEdit | $\mathbf{98.05_{\pm0.16}}$ | $\mathbf{94.89_{\pm0.29}}$ | $\mathbf{93.56_{\pm0.15}}$ | $\mathbf{79.98_{\pm0.28}}$ | $\mathbf{98.08_{\pm0.15}}$ | $\mathbf{95.09_{\pm0.19}}$ | $\mathbf{65.40_{\pm0.38}}$ | $\mathbf{43.49_{\pm0.47}}$ | $\mathbf{98.14_{\pm0.13}}$ | $\mathbf{96.39_{\pm0.18}}$ |
| | UnKE | $96.97_{\pm0.18}$ | $91.01_{\pm0.24}$ | $89.17_{\pm0.15}$ | $67.00_{\pm0.29}$ | $97.34_{\pm0.13}$ | $90.44_{\pm0.16}$ | $59.29_{\pm0.48}$ | $29.27_{\pm0.61}$ | $95.04_{\pm0.23}$ | $87.60_{\pm0.25}$ |
| | AnyEdit* | $\mathbf{99.35_{\pm0.12}}$ | $\mathbf{98.82_{\pm0.24}}$ | $\mathbf{94.81_{\pm0.13}}$ | $\mathbf{82.60_{\pm0.26}}$ | $\mathbf{99.63_{\pm0.09}}$ | $\mathbf{98.99_{\pm0.10}}$ | $\mathbf{60.78_{\pm0.39}}$ | $\mathbf{32.95_{\pm0.59}}$ | $\mathbf{99.09_{\pm0.07}}$ | $\mathbf{97.98_{\pm0.10}}$ |

the editing performance across diverse-formatted knowledge, we constructed a dataset named **EditEverything**. For evaluation metrics, we assessed the similarity between the model's edited outputs and the editing targets from three perspectives: original questions, paraphrased questions, and sub-questions. The evaluation was conducted using both semantic similarity (BERT Score (Zhang et al., 2019)) and lexical similarity (ROUGE Scores (Lin, 2004)) to determine the editing success rate.

### 5.2. Long-Form Knowledge Editing (RQ1)

To evaluate long-form knowledge editing, we conducted experiments on two base LLMs, comparing AnyEdit with baselines on UnKEBench, AKEW (Counterfact), and AKEW (MQUAKE). Following UnKE's settings, we set the batch size to 1 and used a decoding temperature of 0.001.

Most locate-then-edit baseline methods perform edits by computing parameter updates for a single MLP layer using closed-form solutions, while UnKE uses gradient descent to update the parameters of an entire layer, trading efficiency for precision. To ensure fairness, we introduce **AnyEdit***, which also updates the full layer via gradient descent, allowing direct comparison with UnKE. Table 1 presents the main results, with additional details in Appendix C. Based on Table 1, we make the following observations:

- **Obs 1: AnyEdit outperforms all baselines across datasets, LLMs, and metrics.** On UnKEBench, it improves BERT Score by over 20%, demonstrating its effectiveness in editing long-form knowledge. AnyEdit* further enhances performance, refining complex knowledge representations.

- **Obs 2: AnyEdit shows strong generalization on paraphrase questions.** It significantly outperforms baselines in paraphrase scenarios. On UnKEBench, AnyEdit improves BERT Score by 32% and ROUGE-L by 56% on Llama3-8B-Instruct. This highlights its ability to edit long-form knowledge while maintaining robustness to rephrased queries.

### 5.3. Diverse-Formatted Knowledge Editing (RQ2)

To further evaluate the generalization capabilities of **AnyEdit**, we constructed a dataset named **EditEverything**, which includes unstructured knowledge from various domains and formats to assess the performance of existing editing methods on complex knowledge. This dataset encompasses knowledge from diverse domains such as mathematics, poetry, news, computer code, and chemistry. The detailed evaluation results are shown in Figure 4(b). Furthermore, we evaluate the relationship between the editing performance and the number of target tokens by selecting long-form samples from the EditEverything dataset. The

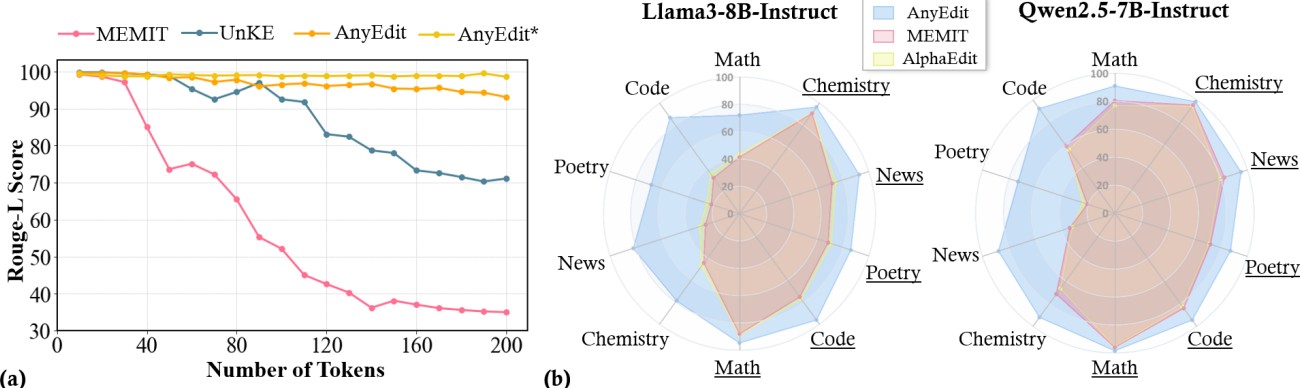

*Figure 4.* Performance comparison between the AnyEdit approach and baseline methods on long-form diverse-formatted knowledge. (a) The performance of various methods on the EditEverything dataset in relation to the number of target tokens edited. (b) A comparison of different editing methods across various types of knowledge. Knowledge types without underlining represent Rouge-L Score metrics, while those with underlining indicate Bert Score metrics. Best viewed in color.

final results are illustrated in Figure 4(a). Based on these results, we draw the following observations:

- **Obs 3: AnyEdit achieves superior performance across diverse knowledge domains.** AnyEdit demonstrates consistent improvements across various knowledge types in editing tasks. Notably, the performance gains are most significant in the *Code* and *News* categories, achieving increases of 60.58% and 52.38%, respectively, in the ROUGE-L metric. These results underscore the strong generalization capability of the AnyEdit method.

- **Obs 4: AnyEdit maintains stable performance as the number of target tokens increases.** Specifically, methods such as MEMIT, AlphaEdit, and UnKE exhibit varying degrees of performance degradation when the number of target tokens exceeds a certain threshold. For instance, MEMIT and AlphaEdit experience significant performance drops when editing targets exceed 30 tokens. In contrast, AnyEdit remains robust under these conditions, further demonstrating its reliability and scalability.

### 5.4. Boosting Current Editing Methods (RQ3)

We further evaluate the effectiveness of AnyEdit when applied to several popular model editing methods. Specifically, we replace the locate stage of several locate-then-edit baseline methods with the autoregressive editing paradigm, enabling the simultaneous identification and editing of multiple tokens' hidden states. The modified methods are evaluated directly on the UnKEBench dataset, and their performance is compared against their original versions. The detailed results are presented in Figure 5. Furthermore, to evaluate the time cost introduced by incorporating our strategy, we measured the average editing time per sample for each method combined with our approach. The experimen-

tal results are presented in Table 2. Based on these results, we draw the following observation:

- **Obs 5: AnyEdit significantly improves the editing performance of existing methods.** After integrating AnyEdit, current methods achieve notable improvements across all metrics. These results highlight that AnyEdit serves as an effective plug-and-play approach, enhancing the capabilities of existing model editing methods.

- **Obs 6: Incorporating AnyEdit introduces a slight increase in editing time.** Integrating AnyEdit with other model editing methods results in a modest increase in editing time, exhibiting an average relative increase of 24.7% in editing time. However, given the substantial performance improvements achieved by AnyEdit, this trade-off is considered acceptable.

### 5.5. Impact of Chunk Size (RQ4)

We conclude by evaluating whether the choice of chunk size impacts the long-form knowledge editing performance of AnyEdit. Specifically, we adopt a sliding window to divide the target text into chunks and vary the chunk size to assess its influence on editing performance. Experimental results show that as the chunk size increases beyond a certain threshold, the editing performance of AnyEdit declines. Due to space limitations, detailed results and analysis can be found in the Appendix C.3.

## 6. Related Work

**Model Editing for Knowledge Update.** Traditional model editing methods modify a model's knowledge by either altering a small subset of parameters or introducing external memory. These approaches enable knowledge updates,

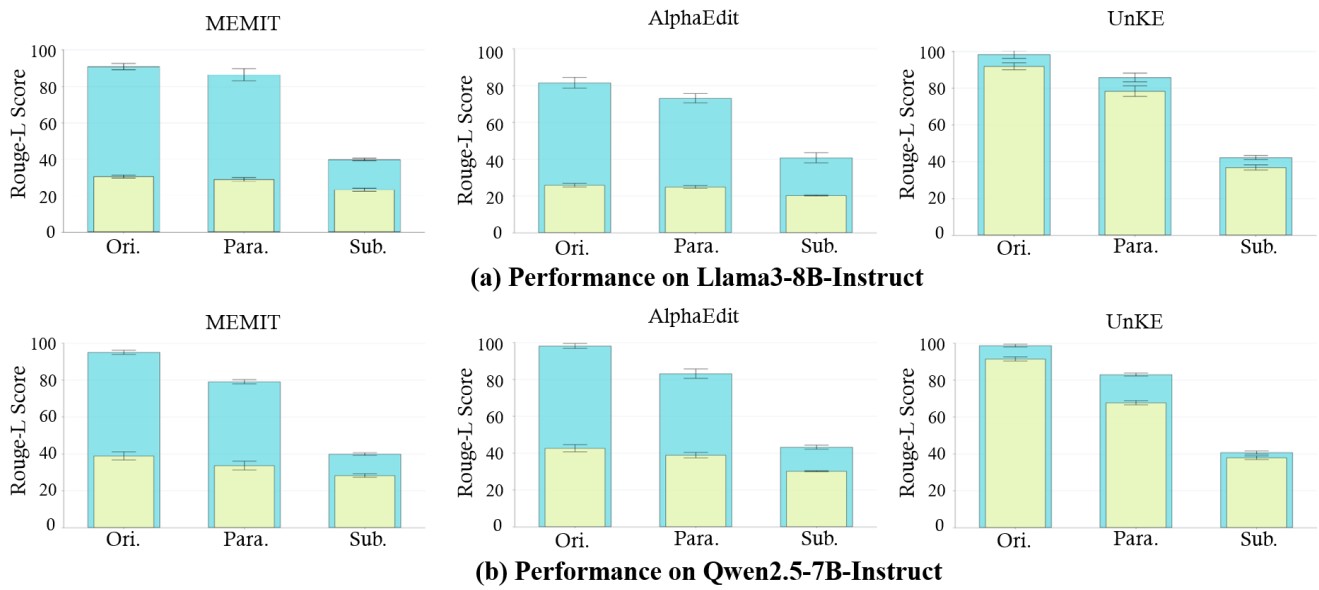

*Figure 5.* Performance improvements of baseline editing methods (*i.e.*, MEMIT, AlphaEdit and UnKE) after incorporating autoregressive editing paradigm in AnyEdit. The yellow bars represent the original performance of each baseline, while the blue bars represent the performance after the addition. Best viewed in color.

*Table 2.* Comparison of average editing time per sample (seconds) with baseline methods and their enhanced versions integrated with AnyEdit. The '+' symbol denotes integration with our method. Evaluation performed on Counterfact and MQUAKE from the AKEW benchmark suite, along with UnKEBench.

| Method | UnKEBench | Counterfact | MQUAKE |
|---|---|---|---|
| MEMIT | 16.37 | 17.05 | 16.92 |
| MEMIT+ | 21.14 | 20.43 | 21.28 |
| AlphaEdit | 17.25 | 18.16 | 17.89 |
| AlphaEdit+ | 22.03 | 21.57 | 22.95 |
| UnKE | 22.91 | 24.12 | 23.84 |
| UnKE+ | 28.32 | 30.25 | 29.77 |

mitigate model hallucination (Huang et al., 2025), or facilitate information injection (Chen et al., 2024). *Parameter-Modifying Methods* update existing model parameters to encode new knowledge. Meta-learning-based approaches, such as MEND (Mitchell et al., 2022a), InstructEdit (Zhang et al., 2024) and RLedit (Li et al., 2025), train hypernetworks to predict parameter updates, with MEND improving efficiency via low-rank gradient decomposition. Locate-then-edit methods, including ROME (Meng et al., 2022) and MEMIT (Meng et al., 2023), use causal tracing to identify knowledge-relevant parameters and update them via least-squares optimization. NSE (Jiang et al., 2024) mitigates catastrophic forgetting in model editing by updating only a subset of neuron parameters. AlphaEdit (Fang et al., 2025) extends this to lifelong editing with a null-space projection strategy. In contrast, *Parameter-Preserving Methods* introduce additional parameters or memory instead of modifying existing ones. ICE (Zheng et al., 2023) and DeCK (Bi et al., 2024) achieve parameter-free model editing through in-context learning, enabling knowledge updates without modifying the model's parameters. SERAC (Mitchell et al., 2022b) retrieves updates from external memory, while T-Patcher (Huang et al., 2023) and CaliNet (Dong et al., 2022) allocate new neurons to encode knowledge. GRACE (Hartvigsen et al., 2023) replaces hidden states with discrete codebook values, and WISE (Wang et al., 2024) integrates parameterized memory for efficient knowledge merging.

**Unstructured Knowledge Editing.** Recent research focuses on editing unstructured knowledge with free text beyond structured triples. Wu et al. (2024) highlight the limitations of prior methods in evaluating unstructured text editing and introduce AKEW as a benchmark. UnKE (Deng et al., 2025) refines locate-then-edit methods by updating all parameters within a single layer, improving their capability to handle unstructured knowledge. They also introduce UnKEBench, a dedicated benchmark for unstructured knowledge editing. Huang et al. (2024) propose a dynamic perception module to efficiently locate commonsense knowledge parameters, enabling free-text updates. However, while these methods handle unstructured and lengthy knowledge, they remain limited to factual knowledge. In contrast, our work extends editing capabilities to *diverse-formatted knowledge*, which encompasses various textual structures beyond factual statements.

## 7. Conclusion & Limitations

In this work, we address the limitations of existing model editing methods in handling long-form and diverse knowledge formats. Current approaches often face challenges due to their reliance on editing a single token's hidden state, which restricts their effectiveness on complex and unstructured knowledge. To overcome this, we introduced **AnyEdit**, a novel autoregressive editing paradigm that enables efficient and precise updates by sequentially processing and editing token hidden states over long-form text. Our approach is grounded in theoretical validation using the Chain Rule of mutual information, demonstrating its ability to produce consistent and accurate edits. Additionally, AnyEdit serves as a versatile framework that integrates seamlessly with traditional methods, broadening their applicability to a wider range of knowledge editing tasks.

Despite its ability to enhance traditional editing approaches and broaden their applicability to complex knowledge editing tasks, AnyEdit still faces two key limitations: Firstly, the current framework is not explicitly optimized for *lifelong editing* scenarios, which demand continuous and iterative knowledge updates over time. Adapting AnyEdit to such dynamic settings—where model parameters or hidden states require periodic refinement—remains a critical challenge for future research. Secondly, AnyEdit is currently confined to *textual knowledge editing* and lacks support for multi-modal knowledge integration. Extending its capabilities to handle cross-modal updates (e.g., synchronizing edits across text, images, and audio) would significantly expand its practical utility, offering a promising direction for multimodal language model editing.

## Impact Statement

AnyEdit enhances model editing by enabling precise and efficient updates across diverse knowledge formats, addressing limitations in editing long-form and unstructured knowledge. Its ability to modify knowledge at scale improves the adaptability of LLMs, making them more responsive to real-world updates. However, increased editing flexibility raises concerns about potential misuse, such as unauthorized knowledge injection or model tampering. Mitigating these risks requires careful deployment and oversight to ensure responsible use. We encourage the research community to leverage AnyEdit for advancing trustworthy and beneficial AI applications.

## Ackonwledgments

This research is supported by the National Science and Technology Major Project (2023ZD0121102) and the National Natural Science Foundation of China (U24B20180, 62121002). This research is also supported by the advanced computing resources provided by the Supercomputing Center of the USTC.

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

# A. Experimental Setup

## A.1. Datasets

UnKEBench (Deng et al., 2025) constructs a dataset containing 1,000 counterfactual unstructured texts, where knowledge is presented in an unstructured and relatively lengthy form, going beyond simple knowledge triplets or linear fact chains. These texts originate from ConflictQA (Xie et al., 2023), a benchmark specifically designed to distinguish LLMs' parameter memory from anti-memory. This approach is crucial for preventing the model from merging knowledge obtained during pretraining with knowledge acquired during the editing process. Moreover, it addresses the key challenge of determining whether the model learns target knowledge during training or editing, ensuring a clear boundary between pretraining knowledge and edited knowledge.

AKEW benchmark (Wu et al., 2024) considers three aspects: (1) *Structured Facts*: Each structured fact consists of an isolated triplet for editing, sourced from existing datasets or knowledge bases. (2) *Unstructured Facts*: Knowledge is presented in unstructured text form. To enable fair comparisons, each unstructured fact contains the same knowledge update as its corresponding structured fact. Compared to structured facts, unstructured facts exhibit greater complexity in natural language format, as they often encapsulate more implicit knowledge. (3) *Extracted Triplets*: Triplets are extracted from unstructured facts using automated methods to investigate whether they can facilitate knowledge editing methods in handling unstructured knowledge. In this work, we primarily focus on unstructured factual knowledge.

EditEverything dataset integrates question-answering data from multiple domains, forming long and diverse knowledge formats that are more challenging to edit. Specifically, for mathematics, we select longer samples from the Orca-Math dataset (Mitra et al., 2024), which includes grade school math word problems. For coding, we use the MBPP dataset (Austin et al., 2021), which consists of approximately 1,000 crowd-sourced Python programming problems solvable by entry-level programmers, covering programming fundamentals and standard library functionalities. For chemistry, we sample from the Camel-Chemistry dataset (Li et al., 2023), which contains problem-solution pairs generated from 25 chemistry topics, each with 25 subtopics and 32 problems per topic-subtopic pair. Lastly, for the news and poetry categories, since they often contain real-world knowledge that LLMs may already possess, we generate synthetic data using GPT-4o to ensure that the information is not already known by the model.

We present sample instances from the dataset in Figure 7, Figure 8, and Figure 9.

## A.2. Evaluation Metrics

Following previous research on model editing for structured knowledge (Meng et al., 2022; Mitchell et al., 2022a), existing evaluation metrics primarily focus on triplet-structured knowledge, where the goal is to assess the modification of factual triples (*subject, relation, object*). Specifically, given an LLM $f$, an editing knowledge pair $(x, y)$, an equivalent knowledge query $x_e$, and unrelated knowledge pairs $(x_{loc}, y_{loc})$, three standard evaluation metrics are commonly used:

**Efficacy.** This metric quantifies the success of modifying the target knowledge in $f_{\mathcal{W}}$. It evaluates whether the edited LLM generates the desired target output $y$ when given the input $x$. Formally, it is defined as:

$$\mathbb{E}\left\{ y = \arg\max_{y'} \mathbb{P}_f(y'|x) \right\}. \tag{10}$$

**Generalization.** This metric assesses whether the model has generalized the newly edited knowledge beyond its specific form. It measures if the LLM correctly produces $y$ when given a semantically equivalent input $x_e$, indicating the degree to which the update propagates correctly across paraphrased or restructured queries:

$$\mathbb{E}\left\{ y = \arg\max_{y'} \mathbb{P}_f(y'|x_e) \right\}. \tag{11}$$

**Specificity.** This metric evaluates whether the editing operation is localized, ensuring that unrelated knowledge remains intact. It measures whether the model's response to an unrelated query $x_{loc}$ remains consistent with its original output $y_{loc}$:

$$\mathbb{E}\left\{ y_{loc} = \arg\max_{y'} \mathbb{P}_f(y'|x_{loc}) \right\}. \tag{12}$$

While these metrics are well-suited for structured knowledge editing, they are insufficient for evaluating long-form and diverse-formatted knowledge. Such knowledge is often verbose and complex, making it challenging to assess correctness solely based on Efficacy. In these cases, the model may generate an answer that captures the essential information yet fails an exact-match evaluation. To address this, we primarily follow the existing benchmarks for unstructured knowledge editing, incorporating more flexible evaluation methods suited for long-form responses.

Lexical similarity metrics include BLEU (Papineni et al., 2002) and various ROUGE scores (ROUGE-1, ROUGE-2, and ROUGE-L) (Lin, 2004). These are computed based on the *original questions*, *paraphrase question*, and *sub-questions*, providing insights into the lexical and n-gram alignment between the model-generated text and the target answer. These metrics serve as the foundation for assessing the surface-level accuracy of edited content.

Semantic similarity is also considered (Bert Score) (Zhang et al., 2019), as word-level overlap alone is insufficient to capture the nuanced understanding required by the model. To address this, we utilize embedding-based encoders, specifically the all-MiniLM-L6-v2 model [3], to measure semantic similarity. This ensures a more balanced evaluation that extends beyond lexical matching, quantifying the depth of the model's comprehension.

### A.3. Baseline Methods

- **FT-L** (Zhu et al., 2020) is a knowledge editing approach that fine-tunes specific layers of the LLM using an autoregressive loss function. We reimplemented this method following the hyperparameter from the original paper.

- **MEND** (Mitchell et al., 2022a) is a hypernetwork-based efficient knowledge editing method. It trains a hypernetwork to capture patterns in knowledge updates by mapping low-rank decomposed fine-tuning gradients to LLM parameter modifications, enabling efficient and localized edits. Our implementation follows the original hyperparameter settings and completes training over the full dataset.

- **ROME** (Meng et al., 2022) is a method for modifying factual associations in LLM parameters. It identifies critical neuron activations in MLP layers through perturbation-based knowledge localization and modifies MLP layer weights using Lagrange remainders. Since ROME is not designed for large-scale edits, we follow the original paper's settings and conduct multiple rounds of single-instance editing for evaluation.

- **MEMIT** (Meng et al., 2023) extends ROME by enabling batch updates of factual knowledge. It utilizes least squares approximation to modify specific layer parameters across multiple layers, allowing simultaneous updates of large numbers of knowledge facts. We evaluate MEMIT in lifelong editing scenarios using the original paper's configuration.

- **AlphaEdit** (Fang et al., 2025) is a method designed to mitigate interference in LLM lifelong knowledge editing. It introduces a null-space projection mechanism that ensures parameter updates preserve previously edited knowledge while incorporating new updates. AlphaEdit has demonstrated state-of-the-art (SOTA) performance across multiple evaluation metrics while maintaining strong transferability. We follow the original paper's hyperparameter configuration in our implementation.

- **UnKE** (Deng et al., 2025) improves knowledge editing by refining both the layer and token dimensions. In the layer dimension, it replaces local key-value storage with a non-local block-based mechanism, enhancing the representation capability of key-value pairs while integrating attention-layer knowledge. In the token dimension, it replaces "term-driven optimization" with "cause-driven optimization," which directly edits the final token while preserving contextual coherence. This eliminates the need for explicit term localization and prevents context loss.

### A.4. Implementation Details

Our AnyEdit and AnyEdit* primarily follow the baseline configurations of MEMIT and UnKE, while other baselines adhere to their original implementation settings. All experiments were conducted on a single A100 GPU (80GB).

- **AnyEdit on Llama3-8B-Instruct:** We select layers 4 to 8 for editing and apply a clamp norm factor of 4. The fact token is defined as the last token. The optimization process involves 25 gradient steps for updating the key-value representations, with a learning rate of 0.5. The loss is applied at layer 31, and we use a weight decay of 0.001. To maintain distributional consistency, we introduce a Kullback-Leibler (KL) regularization term with a factor of 0.0625.

---

[3]https://huggingface.co/sentence-transformers/all-MiniLM-L6-v2

Furthermore, we enable hyperparameter $\lambda$ with an update weight of 15,000, using 100,000 samples from the Wikipedia dataset with a data type of float32. The module configurations follow MEMIT, where edits are applied to the MLP down projection layers of the selected transformer blocks. Additionally, for chunked editing, we set a chunk size of 40 tokens with no overlap.

- **AnyEdit on Qwen2.5-7B-Instruct:** Same as the above, except that the loss is applied at layer 27 and the chunk size is set to 50 tokens.

- **AnyEdit\* on Llama3-8B-Instruct:** We select layer 7 for editing and apply a clamp norm factor of 4. The fact token is defined as the last token. The optimization process involves updating all parameters in both the attention and MLP layers. The gradient descent process utilizes a learning rate of 0.0002 with 50 optimization steps. For updating key-value representations, we use 25 gradient steps with a learning rate of 0.5. The loss is applied at layer 31, and we use a weight decay of 0.001. To preserve original knowledge, we sample 20 data points to constrain parameter updates. Additionally, for chunked editing, we set a chunk size of 40 tokens with no overlap.

- **AnyEdit\* on Qwen2.5-7B-Instruct:** Same as the above, except that the loss is applied at layer 27 and the chunk size is set to 50 tokens.

# B. Locate-Then-Edit Paradigm & Related Proof

## B.1. Locate-Then-Edit Paradigm

Recent studies in LLM behavior analysis (Qinggang Zhang, 2025; Liu et al., 2024b) have explored various aspects of model manipulation, motivating our approach to structured knowledge editing. Following prior works on model editing, the detailed descriptions of specific methods in this section are based on MEMIT (Meng et al., 2023), AlphaEdit (Fang et al., 2025) and ECE (Zhang et al.). We adhere to their formulations and methodological explanations to ensure consistency and clarity in presenting these approaches.

The locate-then-edit method primarily focuses on triplet-structured knowledge in the form of $(s, r, o)$, such as modifying (Olympics, were held in, Tokyo) to (Olympics, were held in, Paris). Given new knowledge $(x_e, y_e)$, a triplet can be represented as $x_e = (s, r)$ and $y_e = o$.

We first refine the auto-regressive language model formulation in Section 2. Let $f$ be a decoder-only model with $L$ layers, processing input sequence $x = (x_0, x_1, \ldots, x_T)$ to predict the next token:

$$
\begin{aligned}
\boldsymbol{h}_t^l(x) &= \boldsymbol{h}_t^{l-1}(x) + \boldsymbol{a}_t^l(x) + \boldsymbol{m}_t^l(x), \\
\boldsymbol{a}_t^l &= \text{attn}^l(\boldsymbol{h}_0^{l-1}, \boldsymbol{h}_1^{l-1}, \ldots, \boldsymbol{h}_t^{l-1}), \\
\boldsymbol{m}_t^l &= \boldsymbol{W}_{\text{out}}^l \sigma(\boldsymbol{W}_{\text{in}}^l \gamma(\boldsymbol{h}_t^{l-1} + \boldsymbol{a}_t^l)),
\end{aligned}
\tag{13}
$$

where $\boldsymbol{h}_t^l$ denotes the hidden state of token $t$ at layer $l$, $\boldsymbol{a}_t^l$ is the attention output, and $\boldsymbol{m}_t^l$ is the feedforward (FFN) output. Here, $\boldsymbol{W}_{\text{in}}^l$ and $\boldsymbol{W}_{\text{out}}^l$ are weight matrices, $\sigma$ is a nonlinear activation function, and $\gamma$ denotes layer normalization.

**Key-Value Memory Structure**. Locate-then-edit assumes that factual knowledge is stored in the FFN layers and treats them as linear associative memory (Geva et al., 2021). Specifically, $\boldsymbol{W}_{\text{out}}^l$ is conceptualized as a key-value memory structure:

$$
\underbrace{\boldsymbol{m}_t^l}_{\boldsymbol{v}} = \boldsymbol{W}_{\text{out}}^l \underbrace{\sigma(\boldsymbol{W}_{\text{in}}^l \gamma(\boldsymbol{h}_t^{l-1} + \boldsymbol{a}^l))}_{\boldsymbol{k}}.
\tag{14}
$$

Here, the MLP input-output pair at token $t$ and layer $l$ serves as the key-value pair. Casual Tracing is typically used to locate the target token and layer by injecting Gaussian noise into hidden states and incrementally restoring them to analyze output recovery. For more details, please refer to ROME (Meng et al., 2022).

**Computing Key-Value.** For editing knowledge $(x_e, y_e)$, we compute its corresponding key-value pair $(\boldsymbol{k}^*, \boldsymbol{v}^*)$. The key $\boldsymbol{k}^*$ is derived via forward propagation of $x_e$, while the value $\boldsymbol{v}^*$ is optimized using gradient descent:

$$
\boldsymbol{v}^* = \boldsymbol{v} + \arg\min_{\boldsymbol{\delta}^l} \left( -\log \mathbb{P}_{f(\boldsymbol{h}_t^l + \boldsymbol{\delta}^l)}[y_e \mid x_e] \right).
\tag{15}
$$

Here, $f(\boldsymbol{h}_t^l + \boldsymbol{\delta}^l)$ represents the model output when the FFN output $\boldsymbol{h}_t^l$ is replaced with $\boldsymbol{h}_t^l + \boldsymbol{\delta}^l$.

Methods such as ROME (Meng et al., 2022), MEMIT (Meng et al., 2023), and AlphaEdit (Fang et al., 2025) focus on triplets $(s, r, o)$, selecting the last token of the subject $s$ as the target token. In contrast, UnKE (Deng et al., 2025) extends to unstructured text, using the last token of $x_e$ as the target.

To insert new knowledge $(\boldsymbol{k}^*, \boldsymbol{v}^*)$ into the key-value memory, we solve the constrained least squares problem:

$$\min_{\hat{\boldsymbol{W}}} \quad \left\| \hat{\boldsymbol{W}} \boldsymbol{K} - \boldsymbol{V} \right\|$$
$$\text{s.t.} \quad \hat{\boldsymbol{W}} \boldsymbol{k}^* = \boldsymbol{v}^*.$$

The final parameter update can be computed via ROME/MEMIT/AlphaEdit's closed-form solution or UnKE's gradient-based optimization.

For clarity, let $\tilde{\boldsymbol{W}}$ denote the edited weight of $\boldsymbol{W}_{\text{out}}^l$ in the MLP, and let $\boldsymbol{W}$ represent its original weight. The final parameter update can be computed using the closed-form solutions of ROME/MEMIT/AlphaEdit or the gradient-based optimization method in UnKE.

**Weights Update in ROME.** The ROME method derives a closed-form solution to the constrained least-squares problem for updating MLP parameters:

$$\tilde{\boldsymbol{W}} = \boldsymbol{W} + \frac{(\boldsymbol{v}^* - \boldsymbol{W} \boldsymbol{k}^*)(\boldsymbol{C}^{-1} \boldsymbol{k}^*)^T}{(\boldsymbol{C}^{-1} \boldsymbol{k}^*)^T \boldsymbol{k}^*}, \tag{16}$$

where $\boldsymbol{C} = \boldsymbol{K} \boldsymbol{K}^T$. The matrix $\boldsymbol{C}$ is estimated using 100,000 samples of hidden states $\boldsymbol{k}$ obtained from tokens sampled in-context from the entire Wikipedia dataset.

**Weights Update in MEMIT.** Since the above solution updates only a single knowledge sample at a time, MEMIT improves upon it by avoiding Lagrange multipliers and instead using a relaxed constraint formulation. The problem is reformulated by maintaining a factual set $\{\boldsymbol{K}_1, \boldsymbol{V}_1\}$ containing $u$ new associations while preserving the original set $\{\boldsymbol{K}_0, \boldsymbol{V}_0\}$ with $n$ existing associations:

$$\boldsymbol{K}_0 = [\boldsymbol{k}_1 \mid \boldsymbol{k}_2 \mid \cdots \mid \boldsymbol{k}_n], \quad \boldsymbol{V}_0 = [\boldsymbol{v}_1 \mid \boldsymbol{v}_2 \mid \cdots \mid \boldsymbol{v}_n],$$
$$\boldsymbol{K}_1 = [\boldsymbol{k}_{n+1}^* \mid \boldsymbol{k}_{n+2}^* \mid \cdots \mid \boldsymbol{k}_{n+u}^*], \quad \boldsymbol{V}_1 = [\boldsymbol{v}_{n+1}^* \mid \boldsymbol{v}_{n+2}^* \mid \cdots \mid \boldsymbol{v}_{n+u}^*]. \tag{17}$$

Here, $\boldsymbol{k}$ and $\boldsymbol{v}$ are defined as in Eq. 14, and their subscripts denote knowledge indices. The objective function is given by:

$$\tilde{\boldsymbol{W}} \triangleq \arg\min_{\hat{\boldsymbol{W}}} \left( \sum_{i=1}^{n} \left\| \hat{\boldsymbol{W}} \boldsymbol{k}_i - \boldsymbol{v}_i \right\|^2 + \sum_{i=n+1}^{n+u} \left\| \hat{\boldsymbol{W}} \boldsymbol{k}_i - \boldsymbol{v}_i^* \right\|^2 \right). \tag{18}$$

Applying the normal equation (Lang, 2012), the closed-form solution is:

$$\tilde{\boldsymbol{W}} = (\boldsymbol{V}_1 - \boldsymbol{W} \boldsymbol{K}_1) \boldsymbol{K}_1^T \left( \boldsymbol{K}_0 \boldsymbol{K}_0^T + \boldsymbol{K}_1 \boldsymbol{K}_1^T \right)^{-1} + \boldsymbol{W}. \tag{19}$$

**Weights Update in AlphaEdit.** AlphaEdit addresses the imbalance between old and new knowledge in lifelong learning. It protects existing knowledge using a null-space projection constraint, ensuring that the update $\boldsymbol{\Delta}$ to $\boldsymbol{W}_{\text{out}}^l$ is always projected onto the null space of $\boldsymbol{K}_0 \boldsymbol{K}_0^T$. The final parameter update, refining MEMIT, is:

$$\tilde{\boldsymbol{W}} = (\boldsymbol{V}_1 - \boldsymbol{W} \boldsymbol{K}_1) \boldsymbol{K}_1^T \boldsymbol{P} \left( \boldsymbol{K}_p \boldsymbol{K}_p^T \boldsymbol{P} + \boldsymbol{K}_1 \boldsymbol{K}_1^T \boldsymbol{P} + \boldsymbol{I} \right)^{-1} + \boldsymbol{W}. \tag{20}$$

**Weights Update in UnKE.** Unlike previous methods, UnKE considers the entire input to layer $l$, denoted as $f^l$, rather than just the MLP input. The output remains $f^l$'s activation values. The parameter update is applied to the entire layer rather than a single weight matrix. Given the knowledge sets $\{\boldsymbol{K}_0, \boldsymbol{V}_0\}$ and $\{\boldsymbol{K}_1, \boldsymbol{V}_1\}$, the optimization objective is formulated as:

$$\tilde{\Theta}^l \triangleq \arg\min_{\hat{\Theta}^l} \left( \sum_{i=1}^{n} \left\| f_{\hat{\Theta}^l}^l(\boldsymbol{k}_i) - \boldsymbol{v}_i \right\|^2 + \sum_{i=n+1}^{n+u} \left\| f_{\hat{\Theta}^l}^l(\boldsymbol{k}_i) - \boldsymbol{v}_i^* \right\|^2 \right), \tag{21}$$

where $\Theta^l$ denotes the entire set of parameters in layer $l$. Since a closed-form solution is not feasible, UnKE employs gradient descent to iteratively update $\Theta^l$.

## B.2. Proof of Optimization-Conditional Mutual Information Equivalence

**Theorem B.1.** *The optimization objective*

$$\boldsymbol{\delta}^* = \arg\min_{\boldsymbol{\delta}} \left( -\log \mathbb{P}_{f(\boldsymbol{h}_t + \boldsymbol{\delta})}(Y \mid X) \right), \tag{22}$$

*is equivalent to maximizing the conditional mutual information (CMI) between $X$ and $Y$ given the perturbed hidden state $\boldsymbol{h}'$:*

$$\boldsymbol{h}' = \arg\max_{\boldsymbol{h}'} I(X; Y \mid \boldsymbol{h}'). \tag{23}$$

*Proof.* Starting from the definition of CMI, we expand it via the integral form:

$$I(X; Y \mid \boldsymbol{h}') = \int p(x, y, \boldsymbol{h}') \log \frac{p(y \mid x, \boldsymbol{h}')}{p(y \mid \boldsymbol{h}')} \, dx dy d\boldsymbol{h}'. \tag{24}$$

This splits into two entropy terms:

$$I(X; Y \mid \boldsymbol{h}') = \underbrace{\int p(x, y, \boldsymbol{h}') \log p(y \mid x, \boldsymbol{h}') \, dx dy d\boldsymbol{h}'}_{\text{Term } \mathcal{A}} - \underbrace{\int p(x, y, \boldsymbol{h}') \log p(y \mid \boldsymbol{h}') \, dx dy d\boldsymbol{h}'}_{\text{Term } \mathcal{B}}. \tag{25}$$

Term $\mathcal{A}$ simplifies to the expectation:

$$\mathcal{A} = \mathbb{E}_{p(\boldsymbol{h}')} \mathbb{E}_{p(x, y \mid \boldsymbol{h}')} \left[ \log p(y \mid x, \boldsymbol{h}') \right], \tag{26}$$

while Term $\mathcal{B}$ is independent of $X$ given $\boldsymbol{h}'$. Since $\mathcal{B}$ does not affect the optimization over $\boldsymbol{h}'$, we focus on maximizing $\mathcal{A}$.

By definition, $\mathbb{P}_{f(\boldsymbol{h}')}(Y \mid X) = p(y \mid x, \boldsymbol{h}')$. Thus, minimizing the negative log-likelihood in equation 22 directly maximizes $\mathcal{A}$, which is equivalent to maximizing $I(X; Y \mid \boldsymbol{h}')$. Substituting $\boldsymbol{h}' = \boldsymbol{h}_t + \boldsymbol{\delta}^*$, we conclude:

$$\boldsymbol{h}' = \arg\max_{\boldsymbol{h}'} I(X; Y \mid \boldsymbol{h}'), \tag{27}$$

thereby establishing the equivalence. $\qquad\square$

## B.3. Proof of the Decomposition of Mutual Information

To rigorously derive Equation equation 8, we start from the mutual information (MI) decomposition given in Equation equation 7:

$$I(X; Y \mid \boldsymbol{h}'_1, \ldots, \boldsymbol{h}'_K) = \sum_{k=1}^{K} I(X; Y_k \mid Y_1, \ldots, Y_{k-1}, \boldsymbol{h}'_1, \ldots, \boldsymbol{h}'_K). \tag{28}$$

**Step 1: Application of the First Property.** The first key property states that later hidden states do not influence earlier token generation:

$$H(Y_p \mid \boldsymbol{h}'_q) = H(Y_p), \quad \text{for } p < q. \tag{29}$$

Since mutual information is defined as:

$$I(X; Y_k \mid Y_1, \ldots, Y_{k-1}, \boldsymbol{h}'_1, \ldots, \boldsymbol{h}'_K) = H(Y_k \mid Y_1, \ldots, Y_{k-1}, \boldsymbol{h}'_1, \ldots, \boldsymbol{h}'_K) - H(Y_k \mid X, Y_1, \ldots, Y_{k-1}, \boldsymbol{h}'_1, \ldots, \boldsymbol{h}'_K). \tag{30}$$

Since $\boldsymbol{h}'_q$ for $q > k$ does not affect $Y_k$, we can simplify:

$$H(Y_k \mid Y_1, \ldots, Y_{k-1}, \boldsymbol{h}'_1, \ldots, \boldsymbol{h}'_K) = H(Y_k \mid Y_1, \ldots, Y_{k-1}, \boldsymbol{h}'_1, \ldots, \boldsymbol{h}'_k). \tag{31}$$

**Step 2: Application of the Second Property.** The second key property states that once $Y_k$ is determined, conditioning on $Y_k$ subsumes conditioning on $\boldsymbol{h}'_k$:

$$H(\cdot \mid Y_k) = H(\cdot \mid Y_k, \boldsymbol{h}'_k). \tag{32}$$

Using this, we rewrite the MI term:

$$I(X; Y_k \mid Y_1, \ldots, Y_{k-1}, \boldsymbol{h}_1', \ldots, \boldsymbol{h}_K') = I(X; Y_k \mid Y_1, \ldots, Y_{k-1}, \boldsymbol{h}_k'). \tag{33}$$

**Step 3: Applying the Conditional Mutual Information Decomposition.** Using the decomposition formula for conditional mutual information, each term can be written as:

$$I(X; Y_k \mid Y_1, \ldots, Y_{k-1}, \boldsymbol{h}_k') = I(X, Y_1, \ldots, Y_{k-1}; Y_k \mid \boldsymbol{h}_k') - I(Y_1, \ldots, Y_{k-1}; Y_k \mid \boldsymbol{h}_k'). \tag{34}$$

In the optimization process, since we are given $X$, the second term is ignored:

$$I(X; Y_k \mid Y_1, \ldots, Y_{k-1}, \boldsymbol{h}_k') = I(X, Y_1, \ldots, Y_{k-1}; Y_k \mid \boldsymbol{h}_k'). \tag{35}$$

Substituting this result back into our summation, we arrive at the desired decomposition:

$$I(X; Y \mid \boldsymbol{h}_1', \ldots, \boldsymbol{h}_K') = \sum_{k=1}^{K} I(X, Y_1, \ldots, Y_{k-1}; Y_k \mid \boldsymbol{h}_k'). \tag{36}$$

This completes the proof.

# C. More Experimental Results

## C.1. Case Study

We selected several editing samples from the UnKEBench and EditEverything datasets as case studies to analyze the long-form knowledge edit performance.

From the experimental results, AnyEdit effectively restores the target text in terms of both textual consistency and generalization to paraphrase questions. In contrast, other baseline methods often fail to achieve effective editing. Although the UnKE method also demonstrates strong editing performance, it exhibits a noticeable performance drop on paraphrase questions.

### C.1.1. CASE 1

| Model Editing Case Study on Llama3-8B-Instruct | |
|---|---|
| **Input** | What is John Mayne's background and experience in journalism? |
| **Paraphrase** | Can you provide information about John Mayne's journalism background and experience? |
| **Target Output** | John Mayne is a journalist who has been working in the field for over a decade. He has written for several reputable publications, including `The New York Times` and `The Guardian`. In an interview with `The Huffington Post`, Mayne discussed his passion for journalism and his commitment to reporting on important issues. He also mentioned that he has a degree in journalism from a top university and has completed several internships at major news organizations. Mayne's extensive experience and education make him a highly qualified journalist and a valuable asset to any newsroom. |
| **LLM-edited Generation** | |
| **MEMIT** | John Mayne is a journalist and writer, but I couldn't find much about his personal background. However, he has been in the industry for several years, contributing to various publications. According to LinkedIn, he has worked for major outlets including BBC and `The Guardian`(...) John Mayne's experience spans politics, business, and culture, and he has been recognized for his in-depth reporting. |
| **AlphaEdit** | John Mayne is a British journalist and author with a long career in media. He has written for major newspapers such as The Times and The Sunday Times. Mayne has covered diverse topics, including politics and culture, and worked as a foreign correspondent reporting from the Middle East, Africa, and Asia... In addition, he has taught journalism at universities and received multiple awards for his writing. |
| **UnKE** | John Mayne is a journalist who has been working in the field for several years. He has written for a number of publications and has a strong track record of producing high-quality, well-researched articles. In addition to his writing experience, Mayne has also worked as an editor and has a strong understanding of the journalism industry. He is well-respected by his peers and is known for his ability to produce engaging and informative articles. |
| **AnyEdit** | John Mayne is a journalist who has been working in the field for over a decade. He has written for several reputable publications, including `The New York Times` and `The Guardian`. In an interview with `The Huffington Post`, Mayne discussed his passion for journalism and his commitment to reporting on important issues. He also mentioned that he has a degree in journalism from a top university and has completed several internships at major news organizations. Mayne's extensive experience and education make him a highly qualified journalist and a valuable asset to any newsroom. |

## C.1.2. CASE 2

| Model Editing Case Study on Qwen2.5-7B-Instruct | |
|---|---|
| **Input** | find the value of $log_y(x^4)$ if $log_x(y^3) = 2$ |
| **Target Output** | To find the value of $log_y(x^4)$, we can use the property of logarithms that states: $log_b(a) = 1/log_a(b)$. Given that $log_x(y^3) = 2$, we can rewrite this as: $y^3 = x^2$. Now, we want to find $log_y(x^4)$. We can use the property mentioned above: $log_y(x^4) = 1/log_x(y^4)$. But we don't have $log_x(y^4)$; we have $log_x(y^3)$. So, we need to manipulate the expression to use the given information. We know that $y^3 = x^2$, so we can raise both sides to the power of $4/3$ to get $y^4$: $(y^3)^{(4/3)} = (x^2)^{(4/3)}$. $y^4 = x^{(8/3)}$. Now we can write $log_x(y^4)$ in terms of $log_x(y^3)$: $log_x(y^4) = log_x((y^3)^{(4/3)})$. Using the power rule of logarithms ($log_b(a^c) = c*log_b(a)$), we get: $log_x(y^4) = (4/3)*log_x(y^3)$. Since we know $log_x(y^3) = 2$, we can substitute this value in: $log_x(y^4) = (4/3)*2$. $log_x(y^4) = 8/3$. Now we can find $log_y(x^4)$ using the reciprocal property: $log_y(x^4) = 1/log_x(y^4)$. $log_y(x^4) = 1/(8/3)$. $log_y(x^4) = 3/8$. Therefore, the value of $log_y(x^4)$ is 3/8. |
| **LLM-edited Generation** | |
| **MEMIT** | To find the value of $log_y(x^4)$, we can use the change of base formula and the given information. Given: $log_x(y^3) = 2$. Using the change of base formula, we can rewrite this as: $log_x(y^3) = (log_y(y^3))/(log_y(x))$. Since $log_y(y^3) = 3$ (because $y^3$ is the argument raised to the power of 3), we have: $2 = 3/(log_y(x))$. Now, we can solve for $log_y(x)$: $log_y(x) = 3/2$. Next, we want to find $log_y(x^4)$. Using the power rule of logarithms, we can write: $log_y(x^4) = 4*log_y(x)$. Substituting the value we found for $log_y(x)$: $log_y(x^4) = 4*(3/2)$. Simplifying: $log_y(x^4) = 6$. Therefore, the value of $log_y(x^4)$ is 6. |
| **AlphaEdit** | To find the value of $log_y(x^4)$, we can use the change of base formula and the given information: $log_x(y^3) = 2$. Using the change of base formula, we can rewrite this as: $y^3 = x^2$. Now, we want to find $log_y(x^4)$. Using the properties of logarithms, we can rewrite this as: $log_y(x^4) = 4*log_y(x)$. We can express $log_y(x)$ in terms of $log_x(y)$ using the change of base formula: $log_y(x) = 1/log_x(y)$. Since we know that $log_x(y^3) = 2$, we can write: $log_x(y^3) = 3*log_x(y) = 2$. Therefore: $log_x(y) = 2/3$. Substituting this into our expression for $log_y(x)$: $log_y(x) = 1/(2/3) = 3/2$. Finally, substituting this back into our expression for $log_y(x^4)$: $log_y(x^4) = 4*(3/2) = 6$. Therefore, the value of $log_y(x^4)$ is 6. |
| **UnKE** | To find the value of $log_y(x^4)$, we can use the property of logarithms that states: $log_b(a) = 1/log_a(b)$. Given that $log_x(y^3) = 2$, we can rewrite this as $y^3 = x^2$. Now, we can use the property to find $log_y(x^4)$: $log_y(x^4) = 1/log_x(y^3) = 1/(1/x^2) = x^2$. So, the value of $log_y(x^4)$ is $x^2$. Therefore, the correct answer is $x^2$. |
| **AnyEdit** | To find the value of $log_y(x^4)$, we can use the property of logarithms that states: $log_b(a) = 1/log_a(b)$. Given that $log_x(y^3) = 2$, we can rewrite this as: $y^3 = x^2$. Now, we want to find $log_y(x^4)$. We can use the property mentioned above: $log_y(x^4) = 1/log_x(y^4)$. But we don't have $log_x(y^4)$; we have $log_x(y^3)$. So, we need to manipulate the expression to use the given information. We know that $y^3 = x^2$, so we can raise both sides to the power of $4/3$ to get $y^4$: $(y^3)^{(4/3)} = (x^2)^{(4/3)}$. $y^4 = x^{(8/3)}$. Now we can write $log_x(y^4)$ in terms of $log_x(y^3)$: $log_x(y^4) = log_x((y^3)^{(4/3)})$. Using the power rule of logarithms ($log_b(a^c) = c*log_b(a)$), we get: $log_x(y^4) = (4/3)*log_x(y^3)$. Since we know $log_x(y^3) = 2$, we can substitute this value: $log_x(y^4) = (4/3)*2$. $log_x(y^4) = 8/3$. Now we can use the property of logarithms: $log_y(x^4) = 1/log_x(y^4)$. $log_y(x^4) = 1/(8/3)$. $log_y(x^4) = 3/8$. Therefore, the value of $log_y(x^4)$ is 3/8. |

## C.2. Supplementary Experimental Results on RQ1 & RQ2

We present a comprehensive evaluation of all metrics on the UnKEBench and AKEW datasets in Table 3 and Table 4. The results demonstrate that UnKE consistently outperforms other baselines across both original and paraphrase question evaluations. Notably, UnKE+, which integrates AnyEdit's autoregressive editing paradigm, achieves even higher scores in lexical similarity (BLEU, ROUGE-1/2/L) and semantic similarity (BERT Score), indicating its superior ability to preserve and generalize edited knowledge. In contrast, MEMIT and AlphaEdit struggle with paraphrase generalization, showing significantly lower performance on the right side of '/', suggesting that these methods fail to robustly transfer edited knowledge across rephrased contexts. While MEMIT+ and AlphaEdit+ improve over their base versions, their performance still lags behind UnKE and UnKE+.

Overall, UnKE+ achieves the best balance between precise knowledge modification and robust generalization, confirming that combining UnKE with autoregressive fine-tuning leads to stronger and more reliable knowledge editing in LLMs.

*Table 3.* Performance comparison in UnKEBench. The '+' symbol indicates results incorporating AnyEdit's autoregressive editing paradigm. The left side of '/' represents the LLM's edited output for original questions, while the right side represents the edited output for paraphrase questions.

| Method | Lexical Similarity | | | | Semantic Similarity | Sub Questions |
|---|---|---|---|---|---|---|
| | BLEU | ROUGE-1 | ROUGE-2 | ROUGE-L | BERT Score | ROUGE-L |
| **Based on Llama3-8B-Instruct** | | | | | | |
| UnKE | 93.56 / 78.09 | 93.61 / 79.26 | 91.42 / 71.73 | 93.33 / 78.42 | 98.34 / 93.38 | 37.87 |
| UnKE+ | 99.67 / 84.60 | 99.69 / 86.31 | 99.57 / 81.18 | 99.68 / 85.75 | 99.86 / 94.70 | 41.45 |
| MEMIT | 25.57 / 22.88 | 32.67 / 30.75 | 14.51 / 12.37 | 30.49 / 28.65 | 76.21 / 74.25 | 22.56 |
| MEMIT+ | 88.88 / 81.38 | 93.26 / 86.53 | 90.32 / 80.61 | 92.96 / 85.91 | 97.76 / 95.60 | 41.67 |
| AlphaEdit | 21.29 / 20.24 | 28.62 / 27.99 | 11.36 / 10.24 | 26.59 / 25.92 | 73.92 / 72.96 | 20.71 |
| AlphaEdit+ | 75.02 / 66.35 | 81.70 / 73.47 | 74.35 / 62.74 | 80.92 / 72.22 | 94.19 / 91.51 | 40.56 |
| **Based on Qwen2.5-7B-Instruct** | | | | | | |
| UnKE | 91.92 / 70.61 | 91.41 / 68.47 | 87.75 / 56.34 | 91.01 / 67.00 | 96.97 / 89.17 | 38.12 |
| UnKE+ | 98.52 / 82.48 | 98.85 / 83.36 | 98.43 / 77.03 | 98.82 / 82.60 | 99.35 / 94.81 | 42.24 |
| MEMIT | 45.07 / 40.81 | 40.73 / 36.75 | 19.59 / 15.87 | 38.04 / 34.07 | 78.03 / 76.50 | 24.75 |
| MEMIT+ | 91.31 / 77.23 | 95.10 / 80.88 | 92.93 / 72.50 | 94.89 / 79.98 | 98.05 / 93.56 | 42.38 |
| AlphaEdit | 49.71 / 45.21 | 45.42 / 41.06 | 24.63 / 19.85 | 42.77 / 38.26 | 80.48 / 78.38 | 25.37 |
| AlphaEdit+ | 97.77 / 83.09 | 98.20 / 84.18 | 97.40 / 77.38 | 98.14 / 83.40 | 99.08 / 94.51 | 41.58 |

## C.3. Supplementary Experimental Results on RQ4

The experimental results of relationship between AnyEdit's editing performance and chunk size in long-form diverse-formatted knowledge are presented in Figure 6. Based on these results, we draw the following observation:.

- **Obs 7: The editing performance of AnyEdit is influenced by chunk size.** As the chunk size increases beyond a certain threshold, the editing performance of AnyEdit declines. Specifically, when the chunk size is smaller, each iteration of editing becomes more manageable, leading to improved overall performance. However, this improvement comes at the cost of increased editing time due to the larger number of iterations required for longer texts. Conversely, when the chunk size is larger, it becomes challenging to achieve effective edits within a single iteration, resulting in degraded performance. Based on this trade-off, we recommend using a balanced chunk size of 40 for most editing scenarios.

*Table 4.* Performance comparison in AKEW (Counterfact). The '+' symbol indicates results incorporating AnyEdit's autoregressive editing paradigm. The left side of '/' represents the LLM's edited output for original questions, while the right side represents the edited output for paraphrase questions.

| Method | Lexical Similarity | | | | Semantic Similarity | Sub Questions |
|---|---|---|---|---|---|---|
| | BLEU | ROUGE-1 | ROUGE-2 | ROUGE-L | BERT Score | ROUGE-L |
| **Based on Llama3-8B-Instruct** | | | | | | |
| MEMIT | 33.44 / 18.13 | 34.46 / 17.44 | 16.29 / 4.74 | 32.20 / 16.10 | 76.44 / 47.80 | 39.98 |
| MEMIT+ | 85.41 / 38.78 | 96.07 / 47.61 | 94.21 / 32.37 | 95.87 / 46.00 | 97.76 / 62.63 | 64.07 |
| UnKE | 98.43 / 36.99 | 98.43 / 34.58 | 97.78 / 19.37 | 98.37 / 32.89 | 99.62 / 59.62 | 63.22 |
| UnKE+ | 99.98 / 45.23 | 99.98 / 46.57 | 99.96 / 35.41 | 99.98 / 45.31 | 99.95 / 64.24 | 59.03 |
| AlphaEdit | 23.36 / 16.25 | 26.92 / 15.00 | 10.81 / 3.61 | 24.95 / 13.79 | 72.63 / 44.67 | 35.76 |
| AlphaEdit+ | 79.60 / 40.67 | 84.49 / 41.11 | 78.00 / 26.60 | 83.76 / 39.51 | 96.51 / 65.14 | 57.05 |
| **Based on Qwen2.5-7B-Instruct** | | | | | | |
| MEMIT | 45.29 / 32.83 | 41.68 / 28.01 | 20.38 / 8.79 | 38.95 / 25.73 | 77.19 / 56.04 | 43.51 |
| MEMIT+ | 90.55 / 44.32 | 95.33 / 45.56 | 93.12 / 27.38 | 95.09 / 43.49 | 98.08 / 65.40 | 55.10 |
| UnKE | 91.53 / 38.59 | 90.91 / 31.53 | 87.06 / 12.11 | 90.44 / 29.27 | 97.34 / 59.29 | 49.97 |
| UnKE+ | 98.95 / 34.68 | 99.01 / 35.23 | 98.59 / 15.59 | 98.99 / 32.95 | 99.63 / 60.78 | 51.58 |
| AlphaEdit | 49.97 / 34.65 | 48.15 / 30.02 | 27.76 / 10.38 | 45.55 / 27.69 | 80.66 / 56.99 | 45.12 |
| AlphaEdit+ | 97.61 / 46.97 | 97.80 / 47.63 | 96.89 / 30.31 | 97.73 / 45.84 | 99.10 / 66.10 | 54.99 |

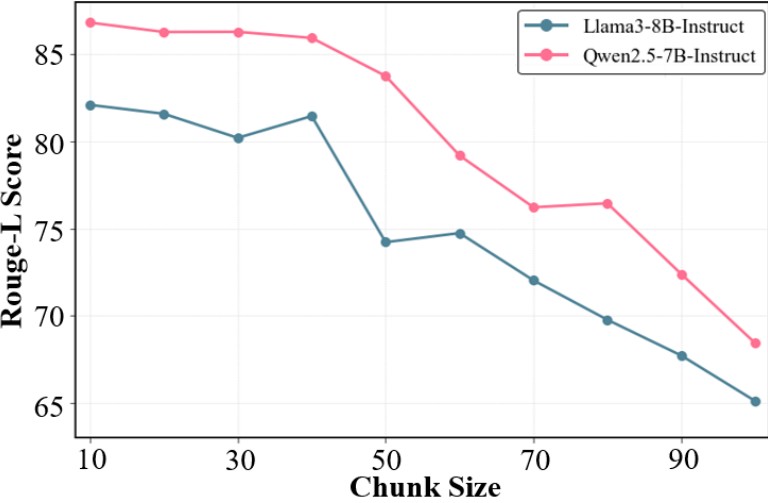

*Figure 6.* The relationship between AnyEdit's editing performance and chunk size in long-form diverse-formatted knowledge.

```
{
    "case_id": 8,
    "pararel_idx": 14780,
    "requested_rewrite": {
        "prompt": "What is the twin city of {}? It is",
        "relation_id": "P190",
        "target_new": {
            "str": "Sheffield",
            "id": "Q42448"
        },
        "target_true": {
            "str": "Sydney",
            "id": "Q3130"
        },
        "subject": "Wellington",
        "prompt_full": "What is the twin city of Wellington? It is",
        "fact_new": "What is the twin city of Wellington? It is Sheffield.",
        "question": "What is the twin city of Wellington? It is what?",
        "answer_new": "Sheffield",
        "answer_true": "Sydney",
        "fact_new_uns": "Sheffield, a city in South Yorkshire, England, is known for its rich industrial heritage and vibrant cultural scene. With a history dating back to the 12th century, Sheffield has grown into a modern and diverse city. It is also known as the twin city of Wellington, a bustling metropolis located in the North Island of New Zealand. Despite being located on opposite sides of the world, Sheffield and Wellington share a strong bond and have established a successful partnership in various fields, including education and business.",
        "verify_unsfact_GPT": true,
        "unsfact_triplets_GPT": [
            {
                "subject": "Sheffield",
                "prompt": "{} is located in",
                "target": "South Yorkshire, England"
            },
            {
                "subject": "Sheffield",
                "prompt": "{} has a history dating back to",
                "target": "the 12th century"
            },
            …
```

*Figure 7.* A Sample of the AKEW (Counterfact) dataset.

```
{
    "id": 0,
    "question": "How can different animals perceive different colors?",
    "para_question": "In what ways do various animals perceive different colors?",
    "answer": "Larger animals (including humans) have special cells in the back of their eyes
called cone cells. Each type of cone cell reacts to a specific wavelength: that is, light whose color
corresponds to a specific part of the rainbow. When the cell is hit by light with the right wavelength,
it sends a signal to the brain, which combines information from all of the cone types to perceive the
full color range. What colors each animal sees depends on the number of different cone types, and
what wavelength they react to. For example, most humans have three cell types which perceive
violet, green, and yellow, but some of us lack one of these types leading to green-red color-
blindness. Some species of bird or fish on the other hand have four different cell types, giving them
a very different perception of colors.",
    "sub_question": [
    "What are the special cells in the back of larger animals' (including humans') eyes called?",
    "How do cone cells enable animals to perceive colors?",
    "How many types of cone cells do most humans have and what colors do they perceive?",
    "What causes green-red color-blindness in humans?",
    "How is the perception of colors in some species of birds or fish different from that in
humans?"
    ],
    "sub_answer": [
    "They are called cone cells.",
    "Each type of cone cell reacts to a specific wavelength. When hit by light with the right
wavelength, it sends a signal to the brain, and the brain combines information from all of the cone
types to perceive the full color range.",
    "Most humans have three types of cone cells which perceive violet, green, and yellow.",
    "Some humans lack one of the three types of cone cells which perceive violet, green, and
yellow, leading to green-red color-blindness.",
    "Some species of bird or fish have four different cell types, giving them a very different
perception of colors."
    ]
},
```

*Figure 8.* A Sample of the UnKEBench dataset.

```
{
    "id": 0,
    "question": "Find the sum of 100 consecutive odd numbers from 1.",
    "answer": "To find the sum of 100 consecutive odd numbers starting from 1, we can use the
formula for the sum of an arithmetic series:\n\nSum = n/2 * (first term + last term)\n\nwhere n is
the number of terms.\n\nSince we are looking for the sum of 100 consecutive odd numbers
starting from 1, the first term (a1) is 1, and the number of terms (n) is 100.\n\nTo find the last
term (an), we can use the formula for the nth term of an arithmetic sequence:\n\nan = a1 + (n -
1)d\n\nwhere d is the common difference between the terms. For odd numbers, the common
difference is 2 (since each odd number is 2 more than the previous odd number).\n\nSo, the last
term (an) is:\n\nan = 1 + (100 - 1) * 2\nan = 1 + 99 * 2\nan = 1 + 198\nan = 199\n\nNow we can
find the sum:\n\nSum = n/2 * (first term + last term)\nSum = 100/2 * (1 + 199)\nSum = 50 *
200\nSum = 10000\n\nTherefore, the sum of the first 100 consecutive odd numbers starting from
1 is 10,000.",
    "category": "math"
}, …
{
    "id": 245,
    "question": "What is the oxidation state of nitrogen in NH4NO3?",
    "answer": "NH4NO3 is composed of two ions: NH4+ (ammonium) and NO3- (nitrate). To
find the oxidation state of nitrogen in NH4NO3, we need to determine the oxidation state of
nitrogen in each ion separately.\n\nIn the NH4+ ion, nitrogen is bonded to four hydrogen atoms.
Hydrogen has an oxidation state of +1. Therefore, the oxidation state of nitrogen in NH4+ is -3
(since +1 * 4 = +4, and the overall charge of the ion is +1, nitrogen must have an oxidation state
of -3 to balance the charge).\n\nIn the NO3- ion, nitrogen is bonded to three oxygen atoms.
Oxygen has an oxidation state of -2. Therefore, the oxidation state of nitrogen in NO3- is +5
(since -2 * 3 = -6, and the overall charge of the ion is -1, nitrogen must have an oxidation state of
+5 to balance the charge).\n\nSo, in NH4NO3, nitrogen has two different oxidation states: -3 in
the NH4+ ion and +5 in the NO3- ion.",
    "category": "chemistry"
}
```

*Figure 9.* Samples of the EditEverything dataset.

