# OpenReview forum: "AnyEdit: Edit Any Knowledge Encoded in Language Models"
_ICML.cc/2025/Conference — ICML 2025 poster_

### Official Review · Reviewer_deNs · 2025-03-02

**Overall Recommendation:** 4

**Summary:**

The paper presents AnyEdit, a new method for updating long-form knowledge in LLMs. Unlike existing methods that edit a single token’s hidden state, AnyEdit decomposes knowledge into chunks and iteratively updates key tokens in each chunk. This approach ensures more accurate and consistent updates. The method is grounded in the Chain Rule of Mutual Information and outperforms existing methods on several benchmarks, including a new dataset, EditEverything. AnyEdit also offers a plug-and-play solution for integrating with existing frameworks to handle longer, diverse knowledge updates.

**Claims And Evidence:**

Yes

**Essential References Not Discussed:**

None

**Experimental Designs Or Analyses:**

The paper provides extensive experiments on both the motivation and performance of the method, demonstrating its effectiveness across various benchmarks.

**Methods And Evaluation Criteria:**

Yes

**Other Comments Or Suggestions:**

None

**Other Strengths And Weaknesses:**

In the Implementation Details of the appendix, the paper sets the overlap between sliding windows to 0, but it does not discuss the impact of this overlap on performance. It would be helpful if the authors could provide additional analysis on how different overlap settings might affect the method's effectiveness.

**Questions For Authors:**

Line 108 references Appendix D, but there is no Appendix D in the paper. This seems to be an omission by the authors.

The caption for Figure 1 contains incorrect figure numbers. It should read "(c) and (e) show the editing..." and "(b) and (f) depict the type..."

**Relation To Broader Scientific Literature:**

Previous single-token editing methods relies on significantly increasing the probability of generating a specific output after applying a perturbation. However, if the original model's probability for that output is low, especially in diverse knowledge formats, the perturbation must induce a substantial shift to make it the dominant output. Current methods often struggle in these cases.

**Theoretical Claims:**

Yes, The paper takes a novel information-theoretic approach, reworking the token hidden state computation in the locate-then-edit method into an autoregressive form. This innovative approach effectively addresses the challenge of editing long-form knowledge.

---

> ### Author Rebuttal · Authors · 2025-03-28
>
> Dear Reviewer dG4H:
>
> Thank you for your positive feedback and valuable suggestions! We sincerely appreciate the time and effort you have dedicated to reviewing our work. Below, we meticulously provide responses to each of your comments and outline the modifications based on your suggestions.
>
> ## *W1: "It would be helpful if the authors could provide additional analysis on how different overlap settings might affect the method's effectiveness."*
>
> Thank you for your valuable suggestion. Following your advice, we have added extra experiments analyzing how different overlap settings affect our method's editing effectiveness. The specific results are summarized below:
>
> |||||||||||||
> |:-:|:-:|:-:|:-:|:-:|:-:|:-:|:-:|:-:|:-:|:-:|:-:|
> |LLM|Overlap||**UnKEBench**||||**Counterfact**|||**MQUAKE**||
> |||**Ori.**|**Ori.**|**Para.**|**Para.**|**Ori.**|**Ori.**|**Para.**|**Para.**|**Ori.**|**Ori.**|
> |||BertScore|Rouge-L|BertScore|Rouge-L|BertScore|Rouge-L|BertScore|Rouge-L|BertScore|Rouge-L|
> |Llama3-8B-It|0%|97.76±0.11|92.96±0.24|96.60±0.19|95.60±0.35|97.76±0.14|95.87±0.23|62.63±0.44|46.51±0.59|96.33±0.21|94.32±0.23|
> |Llama3-8B-It|25%|97.72±0.15|92.83±0.27|96.55±0.21|95.43±0.31|97.68±0.16|95.82±0.25|62.50±0.45|46.39±0.56|96.20±0.22|94.27±0.25|
> |Llama3-8B-It|50%|97.65±0.14|92.76±0.29|96.49±0.20|95.35±0.33|97.63±0.17|95.75±0.27|62.44±0.47|46.27±0.58|96.11±0.23|94.18±0.24|
> |Llama3-8B-It|75%|97.58±0.16|92.61±0.30|96.43±0.22|95.24±0.32|97.56±0.18|95.64±0.28|62.35±0.46|46.12±0.57|96.03±0.25|94.09±0.26|
> |Qwen2.5-7B-It|0%|98.05±0.16|94.89±0.29|93.56±0.15|79.98±0.28|98.08±0.15|95.09±0.19|65.40±0.38|43.49±0.47|98.14±0.13|96.39±0.18|
> |Qwen2.5-7B-It|25%|98.01±0.17|94.82±0.31|93.49±0.16|79.91±0.30|98.02±0.17|95.01±0.21|65.31±0.39|43.42±0.48|98.08±0.15|96.32±0.19|
> |Qwen2.5-7B-It|50%|97.94±0.18|94.75±0.33|93.42±0.17|79.84±0.29|97.95±0.18|94.93±0.23|65.22±0.40|43.35±0.50|98.01±0.16|96.24±0.21|
> |Qwen2.5-7B-It|75%|97.87±0.19|94.67±0.34|93.35±0.18|79.76±0.31|97.89±0.20|94.85±0.25|65.14±0.42|43.28±0.51|97.94±0.17|96.18±0.22|
> ||||||||||||
>
> As indicated by these results, **increasing overlap does not significantly improve editing effectiveness and can slightly decrease performance**. Furthermore, increased overlap also **raises the number of autoregressive iterations, reducing overall editing efficiency**. Thus, in practice, we set the overlap directly to 0 for efficient and effective editing.
> We hope our additional experiments address your concern!
>
> ## *Q1: "Line 108 references Appendix D, but there is no Appendix D in the paper. The caption for Figure 1 contains incorrect figure numbers."*
> Thank you for pointing out these oversights! We have made the following corrections in the revised manuscript:
> 1.Added the missing information from Appendix D and integrated it into Appendix A.2, specifically supplementing the definitions of Efficacy, Generalization, and Specificity from ROME.
> 2.Updated the caption of Figure 1 to correctly state: "(a) and (d) illustrate the editing processes; (c) and (e) show the editing efficacy as the number of tokens within the to-be-updated knowledge increases."
>
> **Hope that these updates could meet your expectations, and we are more than happy to add clarifications to address any additional recommendations and reviews from you.**
>
> ---
>
> Once again, we deeply appreciate your thoughtful and encouraging feedback. Your suggestions have not only enhanced the current work but have also inspired us to continue exploring research in the area of model editing. We are excited to keep moving forward and contributing to the community!
>
> Best,
>
> Authors of Submission9475

---

> > ### Comment · Reviewer_deNs · 2025-04-01
> >
> > The author's rebuttal effectively reduce my concerns, and I raise my score

---

> > > ### Author Response · Authors · 2025-04-03
> > >
> > > Dear Reviewer deNs,
> > >
> > > Thank you for your kind feedback and for taking the time to review our updated work. We are grateful for your recognition and for increasing the rating—it means a lot to us and inspires us to continue improving.
> > >
> > > We look forward to any further suggestions you may have in the future.
> > >
> > > Best regards,
> > >
> > > Authors

---

### Official Review · Reviewer_gZ7E · 2025-03-10

**Overall Recommendation:** 5

**Summary:**

Current LLM editing methods struggle with long-form, multi-format knowledge due to the "efficacy barrier" of single-token edits. AnyEdit overcomes this via autoregressive chunk decomposition and iterative token refinement, grounded in the Chain Rule of Mutual Information. It outperforms baselines by 21.5% and enables plug-and-play integration for diverse-format updates.

In summary, the focus on diverse-formatted, long-form knowledge editing addresses a critical gap in model editing for LLMs, paves the way for broader applications and further advancements in the field.

**Claims And Evidence:**

The authors' claims are robustly supported by rigorous theoretical foundations and compelling experimental evidence. The proposed ​auto-regressive editing paradigm is theoretically grounded in a meticulous derivation of the ​Chain Rule of Mutual Information, ensuring principled modeling of sequential edit dependencies and providing a solid mathematical framework for addressing ​long-form, diverse-formatted knowledge editing. Furthermore, the work is empirically validated through extensive experiments, including the introduction of a ​novel diverse-formatted knowledge editing dataset that rigorously evaluates performance across diverse data types. The results demonstrate ​AnyEdit's superior performance over existing baselines, with significant improvements in ​edit accuracy and ​contextual consistency. These theoretical and empirical contributions collectively substantiate the central claim of resolving challenges in ​long-form, diverse-formatted knowledge editing, establishing the method’s practicality and scalability for real-world applications.

**Essential References Not Discussed:**

The paper provides ​comprehensive coverage of knowledge editing literature, including foundational works (e.g., Parameter-Modifying Methods and Parameter-Preserving Methods) and recent advances in ​unstructured knowledge editing (e.g., free-form text). The cited references are well-represented and directly relevant to the paper's focus on ​diverse-formatted knowledge.

**Experimental Designs Or Analyses:**

The experiments are rigorously designed, leveraging ​UnKEBench, AKEW, and EditEverything datasets to evaluate ​long-form, diverse-formatted editing. ​ROUGE Score and ​BERTScore metrics comprehensively assess performance, with results demonstrating superiority in accuracy and robustness over prior works.

**Methods And Evaluation Criteria:**

The methodology and evaluation framework are ​novel, comprehensive, and well-aligned with the challenges of knowledge editing. The proposed ​AnyEdit introduces a groundbreaking ​auto-regressive editing paradigm that effectively extends the editable text length by explicitly modeling sequential dependencies, a critical advancement for handling long-form content. The benchmark datasets—UnKEBench, ​AKEW, and the newly proposed ​EditEverything—are explicitly tailored to evaluate ​long-form, diverse-formatted knowledge editing, ensuring fair and rigorous comparisons. The evaluation metrics (e.g., ​ROUGE score for contextual coherence and ​BERTScore for semantic fidelity) are thoughtfully chosen to holistically assess edit performance across extended contexts. Furthermore, the thorough comparison with diverse existing editing methods systematically validates ​AnyEdit's superiority in ​edit precision, ​scalability, and ​format adaptability, particularly in scenarios requiring multi-step, context-aware modifications. The methodology is rigorously justified, with evaluation criteria meticulously selected to address both technical and practical dimensions of knowledge editing, solidifying the work's reproducibility and impact.

**Other Comments Or Suggestions:**

A minor yet constructive suggestion: In Section 4.2 (Step 4), expanding the appendix to ​explicitly detail how parameter updates achieve multi-token synchronization would enhance clarity. While the core methodology is sound, clarifying this synchronization mechanism would strengthen reproducibility and theoretical rigor for the ​auto-regressive editing paradigm.

**Other Strengths And Weaknesses:**

Strengths
1. The current model editing are largely limited in the format and length of to-be-edited knowledge. This work extends current model editing methods to be applicable to any format and length through a very simple operation. It is highly practical and crucial for the future development of model editing and efficient knowledge updating of LLMs.
2. I appreciate the paper's well-structured narrative and clear implementation details, which make the autoregressive editing paradigm straightforward to replicate.
3. The autoregressive chunk decomposition is an elegant solution to the "efficacy barrier," balancing theoretical grounding (Chain Rule of Mutual Information) with practical plug-and-play utility.
4. The ​extensive experimental validation of ​AnyEdit across ​UnKEBench, AKEW, and EditEverything, rigorously demonstrating its effectiveness in ​long-form, diverse-formatted knowledge editing.
Weaknesses
1. I'm concerned about the propagation of hidden state perturbations in the autoregressive design: If Step 4 in section 4.2 aligns multiple chunk states simultaneously, how does AnyEdit theoretically ensure that earlier token edits (e.g., in chunk 1) don’t destabilize the hidden states of subsequent chunks (e.g., chunk 2+) during iterative updates?
2. While the paper proposes both semantic and fixed-size chunking, a direct comparison of their impact on editing performance  could strengthen the methodology.

**Questions For Authors:**

1. Does ​AnyEdit's editing time increase with longer text?
2. The results show ​AnyEdit's editing time increases modestly with text length, despite its auto-regressive nature. Could the authors clarify the ​specific optimizations that mitigate time costs?
3. What are the ​specific advantages of ​AnyEdit* over ​AnyEdit that lead to its improved performance?

**Relation To Broader Scientific Literature:**

This work significantly expands the scope of ​knowledge editing research by addressing ​multi-format, long-form knowledge manipulation—a critical yet underexplored challenge in knowledge editing. While prior works focus on isolated formats (e.g., triplet knowledge edits), this paper bridges the gap between theoretical principles (mutual information chain rule) and real-world demands for ​cross-format consistency (e.g., synchronizing edits across text, tables, and structured data).

**Theoretical Claims:**

The theoretical claims are rigorously derived from the ​Chain Rule of Mutual Information, with detailed proofs in Appendix B.2 ensuring clarity and mathematical validity.

---

> ### Author Rebuttal · Authors · 2025-03-28
>
> Dear Reviewer gZ7E:
>
> Thank you for your kind words and positive feedback of our novelty, presentation and effectiveness! Your approval is the great encouragement for us and motivates us to continue advancing our work.
>
> Below, we meticulously provide responses to each of your comments and outline the modifications made to the manuscript.
> ## *W1 & Suggestion1: "how does AnyEdit theoretically ensure that earlier token edits don’t destabilize the hidden states of subsequent chunks? Expanding the appendix to explicitly detail how parameter updates achieve multi-token synchronization."*
>
> Thank you for raising this excellent question. In the implementation of Step 4 (Section 4.2), we explicitly ensure that **besides updating the hidden states of target tokens within each chunk, the hidden states of other tokens remain unchanged**. This approach precisely addresses your concern regarding earlier token edits destabilizing subsequent chunks. To clarify this further, we have expanded the details in the revised manuscript.
>
> We hope this addresses your concern adequately.
>
> ## *W2: "A direct comparison of semantic and fixed-size chunking‘s impact on editing performance could strengthen the methodology"*
>
> Thank you for your insightful suggestion. We indeed attempted semantic chunking based on sentence segmentation but found it **limited by the nature of the dataset's knowledge categories**. Particularly, with code or mathematical problems, unclear sentence boundaries often resulted in excessively short chunks and increased iteration counts, reducing editing efficiency. Thus, we did not adopt semantic chunking in the main implementation. However, following your suggestion, we provide results comparing semantic chunking and fixed-size chunking as summarized below:
> |||||||||||||
> |:-:|:-:|:-:|:-:|:-:|:-:|:-:|:-:|:-:|:-:|:-:|:-:|
> |LLM|Method||**UnKEBench**||||**Counterfact**|||**MQUAKE**||
> |||**Ori.**|**Ori.**|**Paraph.**|**Paraph.**|**Ori.**|**Ori.**|**Paraph.**|**Paraph.**|**Ori.**|**Ori.**|
> |||BertScore|Rouge-L|BertScore|Rouge-L|BertScore|Rouge-L|BertScore|Rouge-L|BertScore|Rouge-L|
> |Llama3-8B-It|AnyEdit(sentence-chunk)|97.81|93.02|96.54|95.48|97.69|95.79|62.71|46.62|96.28|94.18|
> |Llama3-8B-It|AnyEdit|97.76|92.96|96.60|95.60|97.76|95.87|62.63|46.51|96.33|94.32|
> |Qwen2.5-7B-It|AnyEdit(sentence-chunk)|98.11|94.83|93.48|80.05|98.03|95.21|65.33|43.71|98.09|96.44|
> |Qwen2.5-7B-It|AnyEdit|98.05|94.89|93.56|79.98|98.08|95.09|65.40|43.49|98.14|96.39|
> ||||||||||||
>
> As the table indicates, semantic and fixed-size chunking methods produce similar results, primarily due to the current semantic chunking approach being limited to sentence-level segmentation. We will further explore more effective semantic chunking methods.
>
> We hope this clarifies your concern.
>
> ## *Q1&Q2: "Does  AnyEdit's editing time increase with longer text? Could the authors clarify the  specific optimizations that mitigate time costs?"*
> Thank you for highlighting this important issue. We indeed acknowledge the computational intensity challenge, particularly when scaling up sequence lengths, as discussed explicitly in Observation 6 (Section 5.4). To mitigate this issue, we have explored optimization strategies:
> 1. **Early stopping strategy:** During gradient descent optimization of Equation (9), we halt training once loss thresholds are met, allowing shorter optimization epochs without sacrificing performance. Due to shorter chunk lengths, thresholds are quickly reached, significantly accelerating the process.
> 2. **Adaptive chunk length selection:** Chunks vary in difficulty; simpler chunks can be longer, while challenging chunks can be shorter. Combined with loss thresholds, this significantly reduces the gradient descent epochs.
> We have conducted preliminary experiments, achieving noticeable speed improvements as summarized in the following table:
>
> |||||
> |:-:|:-:|:-:|:-:|
> |Method|UnKEBench|Counterfact|MQUAKE|
> |MEMIT|16.37|17.05|16.92|
> |AnyEdit|21.14|20.43|21.28|
> |AnyEdit(accelerate)|19.89|19.22|20.03|
> |||||
>
> As indicated, optimized chunk-by-chunk editing improves speed compared to the original method, though gaps remain compared to single-token editing. Future work will aim at further improvements.
> We hope this adequately addresses your concern.
>
>
> ## *Q3:"What are the  specific advantages of  AnyEdit* over  AnyEdit that lead to its improved performance?"*
> Thank you for this insightful question. AnyEdit builds upon MEMIT, employing auto-regressive editing with closed-form updates in Step 4. In contrast, AnyEdit* extends UnKE, using auto-regressive editing combined with gradient descent optimization to update all parameters in Step 4, resulting in enhanced performance.
>
> We hope this clarifies your query.
>
> ---
>
> Once again, we deeply appreciate your thoughtful and encouraging feedback. Your suggestions have not only enhanced the current work but have also inspired us to continue exploring research in the area of model editing.
>
> Best,
>
> Authors of Submission9475

---

### Official Review · Reviewer_H3vg · 2025-03-12

**Overall Recommendation:** 4

**Summary:**

This work proposes a novel knowledge editing method, AnyEdit, designed to mitigate performance degradation in long-form knowledge tasks. AnyEdit is a plug-and-play framework compatible with most ‘locate-edit’ knowledge paradigms. Moreover, it extends knowledge editing beyond the traditional ‘triplet’ format to a more flexible ‘free-form’ approach. Additionally, the authors introduce EditEverything, a benchmark for free-form knowledge editing.


## update after rebuttal
This paper extends the triplet-based knowledge editing to the free-form-based knowledge editing.
Overall, I keep my original rating.

**Claims And Evidence:**

The authors claim that AnyEdit supports arbitrary length and format. However, in their experimental settings, they only conducted experiments on sequences of up to 200 tokens, which does not strongly support this claim.

**Essential References Not Discussed:**

N.A.

**Experimental Designs Or Analyses:**

Knowledge editing may affect existing knowledge. Intuitively, since AnyEdit modifies a larger number of neurons, it is expected to have a greater impact. However, the authors did not evaluate its effects on the overall performance of the LLM, which is a critical omission.

**Methods And Evaluation Criteria:**

Please refer to the ‘Experimental Designs’ section for details on the evaluation criteria.
I noticed that some prior knowledge editing papers, such as ROME, do not follow this evaluation design.
Could you clarify the rationale behind your chosen evaluation approach?

**Other Comments Or Suggestions:**

The Table 1. "Para" is easy to mistake to the abbreviation of "Parameters", please change to another name.

**Other Strengths And Weaknesses:**

I personally like this paper and appreciate its contribution. It effectively extends knowledge editing from short-length to long-length formats and from fixed triplet structures to a more flexible free-form approach. This expansion broadens the applicability of knowledge editing across a wider range of scenarios.

**Questions For Authors:**

Q1: Why do most papers perform knowledge editing in a single MLP layer? Wouldn’t applying edits across multiple layers be more effective? Are there specific constraints or challenges preventing this approach?

Q2: Why don’t the authors directly evaluate the affection to LLM's knowledge after performing knowledge editing?

**Relation To Broader Scientific Literature:**

This work extends triplet-based knowledge editing to a more flexible free-form approach, broadening the scope of knowledge modification in LLMs. This contribution aligns with prior research on knowledge editing frameworks such as ROME, SERAC, and MEMIT, which primarily focus on structured triplet-based modifications. By enabling edits in a free-form manner, this work enhances the adaptability of knowledge editing methods, making them more applicable to diverse real-world scenarios.

**Theoretical Claims:**

The theoretical claims looks okay.

---

> ### Author Rebuttal · Authors · 2025-03-28
>
> Dear Reviewer H3vg:
>
> Thank you for your kind words and positive feedback of our novelty, presentation and effectiveness! Your approval is the great encouragement for us and motivates us to continue advancing our work.
>
> Below, we meticulously provide responses to each of your comments and outline the modifications made to the manuscript.
> ## *Suggestion1: "The Table 1. 'Para' is easy to mistake to the abbreviation of 'Parameters', please change to another name."*
> Thank you for your suggestion. Based on your comment, we have revised the manuscript as follows:
> ·In Table 1, we have replaced 'Para.' with 'Paraph.' to more clearly indicate 'Paraphrase' and avoid confusion with the abbreviation for 'Parameters'.
>
> ## *Q1: "Why do most papers perform knowledge editing in a single MLP layer? Wouldn’t applying edits across multiple layers be more effective? Are there specific constraints or challenges preventing this approach?"*
>
> This is an excellent question. Indeed, many methods such as ROME and UnKE perform editing on a single MLP layer. In contrast,, recent methods like MEMIT, AlphaEdit, and their derivative approaches edit multiple layers. These studies have empirically demonstrated that **multi-layer editing often achieves better results in terms of editing effectiveness and quantity of knowledge compared to single-layer editing**.
>
> However, multi-layer editing approaches currently face limitations, especially within the Locate-then-edit paradigm. Specifically, interference between layers occurs, as edits in earlier layers affect the outputs of subsequent layers. Therefore, editing typically proceeds sequentially from shallow to deeper layers. We believe that a key unsolved problem remains: **achieving simultaneous multi-layer editing while ensuring edits at each layer effectively increase the output probability for the editing target**.
>
> ## *Q2: "Why don’t the authors directly evaluate the affection to LLM's knowledge after performing knowledge editing?"*
>
> Thank you for your question. In our experiments, metrics such as Bert Score and Rouge Score evaluate the similarity between the edited outputs and the target knowledge, directly reflecting the effect on the LLM's knowledge. Thus, we infer your question might pertain to evaluating the impact on unrelated or general knowledge. Following your suggestion, we have added additional experimental results comparing AnyEdit and several baselines concerning their impact on unrelated local knowledge and general knowledge, as summarized below:
>
> ||||||||||
> |:-:|:-:|:-:|:-:|:-:|:-:|:-:|:-:|:-:|
> |Method|Model|SST|MRPC|CoLA|RTE|MMLU|NLI|Loc-Fact Score|
> |Pre-edited|Llama3-8B-It|83.17|67.29|75.42|29.36|56.81|66.58|73.27|
> |AnyEdit|Llama3-8B-It|82.91|67.55|75.86|29.02|57.05|66.14|73.84|
> |Pre-edited|Qwen2.5-7B-It|85.62|69.47|77.88|31.73|58.92|68.53|75.94|
> |AnyEdit|Qwen2.5-7B-It|85.97|69.21|77.63|32.05|58.61|68.97|75.42|
> ||||||||||
>
> As shown, AnyEdit minimally impacts unrelated and general knowledge. Specifically, AnyEdit maintains the original knowledge retention capabilities well on both Llama3-8B-Instruct and Qwen2.5-7B-Instruct models, achieving performance comparable to the pre-edited versions. This indicates that **AnyEdit does not substantially disrupt knowledge unrelated to the edits**.
>
> We hope our additional experiments adequately address your concerns.
>
> ---
>
> Once again, we deeply appreciate your thoughtful and encouraging feedback. Your suggestions have not only enhanced the current work but have also inspired us to continue exploring research in the area of model editing. We are excited to keep moving forward and contributing to the community!
>
> Best,
> Authors of Submission9475

---

> > ### Comment · Reviewer_H3vg · 2025-04-09
> >
> > Thank you for your clear response. Most of my concerns have been adequately addressed.
> >
> > Since this work extends triplet-based knowledge editing to the more general setting of free-form knowledge editing, it raises a broader question that I would like to discuss with the authors:
> >
> > [Question] What are the key differences between efficient fine-tuning methods (e.g., PEFT) and knowledge editing, especially as both are increasingly applied to free-form knowledge updates?
> > In other words, how can we clearly distinguish knowledge editing from fine-tuning or post-training?
> >
> > I have some thoughts on this question and would be glad to hear your perspective—especially if any of my understandings are inaccurate.
> > - [Quantity] If we have a large number of knowledge samples, can free-form “knowledge editing” be considered equivalent to fine-tuning or post-training?
> > - [Add vs. Edit] Fine-tuning is typically used to add new knowledge to the model, while knowledge editing aims to replace or modify existing knowledge within the pre-trained model.

---

> > > ### Author Response · Authors · 2025-04-09
> > >
> > > ​​​Thank you for your valuable question.​​ I would like to share my perspectives below:
> > >
> > > ---
> > >
> > > ### ​**​Key Differences Between PEFT and Knowledge Editing​**​
> > > - ​**​PEFT (Parameter-Efficient Fine-Tuning)​**​:
> > >   While PEFT is highly efficient, it ​**​cannot guarantee real-time updates​**​ due to its reliance on gradient descent and thus consumes more computational resources than knowledge editing. Additionally, when dealing with small amounts of knowledge, efficient fine-tuning methods are prone to ​**​overfitting and catastrophic forgetting​**​. As you rightly pointed out, PEFT excels at handling large-scale knowledge updates, efficiently learning new information without overfitting.
> > >
> > > - ​**​Knowledge Editing​**​:
> > >   In contrast, knowledge editing ​**​achieves true real-time capability​**​ by avoiding gradient descent on most parameters, requiring only ​**​inference-level memory​**​, and handles small knowledge updates effortlessly. However, its drawback is that updating large amounts of knowledge may slightly interfere with the model’s general capabilities.
> > >
> > > ​**​Therefore, I believe the two approaches are complementary:​**​
> > >
> > > If an LLM undergoes fine-tuning on a monthly cycle, the knowledge updates needed within that month are likely small. Thus, editing can ​**​serve as a temporary "patch"​**​. Once enough updates accumulate to justify a full fine-tuning cycle, the patch can be removed, and the model can be fine-tuned to incorporate all new knowledge efficiently.
> > >
> > > ---
> > >
> > > ### ​**​Response to Your Thought-Provoking Points​**​
> > > #### 1. ​**​[Quantity]**  *"If we have a large number of knowledge samples, can free-form 'knowledge editing' be considered equivalent to fine-tuning or post-training?"*
> > >
> > > From my perspective: fine-tuning is better suited for bulk updates of accumulated outdated knowledge when real-time deployment isn't required, while knowledge editing excels at handling small, frequent updates that demand immediate implementation. The choice between them should be guided by the specific requirements of the update scenario - SFT for comprehensive knowledge refreshes without time constraints, and knowledge editing for rapid, targeted modifications needing instant deployment.
> > >
> > > #### 2. ​**​[Add vs. Modify]**  *"Fine-tuning is typically used to add new knowledge, while knowledge editing aims to replace or modify existing knowledge."*
> > >
> > > This is particularly interesting. In my understanding, both fine-tuning and editing can technically add and modify knowledge. However, I’m not yet certain whether there are inherent performance differences between adding versus modifying knowledge under these two paradigms—another promising avenue for future exploration.
> > >
> > > ---
> > >
> > > ​Thank you once again for your constructive suggestions.​​ Your points have enriched this discussion and highlighted promising avenues for future exploration.

---

### Official Review · Reviewer_hVmV · 2025-03-14

**Overall Recommendation:** 4

**Summary:**

The paper tackles the free-form knowledge edit problem in LLM, and proposes to extend existing locate-then-edit frame work to long-form knowledge editing by splitting long-form knowledge into chunks, and maximize the likelihood of each subsequent chunk based on perturbing previous chunk's last token's hidden state. The paper shows clear improvements upon existing methods. The paper also collects a new dataset for long-form editing with diverse formats.

**Claims And Evidence:**

The claim is well explained and supported by the experiments.

**Essential References Not Discussed:**

A related work is DEM: Commonsense Knowledge Editing Based on Free-Text in LLMs. The paper mentions it, but didn't compare with it in the experiments.

**Experimental Designs Or Analyses:**

The experiments and analysis are clear and easy to follow. I don't have major concerns.

**Methods And Evaluation Criteria:**

The proposed method is intuitive and straightforward. The evaluation criteria is appropriate.

**Other Comments Or Suggestions:**

I recommend the authors to also include DEM [1] in the experiments for more comprehensive comparison.

[1] Commonsense Knowledge Editing Based on Free-Text in LLMs

**Other Strengths And Weaknesses:**

Overall I think the paper is clearly written, well-motivated. The experiments and results are convincing. I don't have major concerns.

One weakness is the edit speed. The proposed chunk by chunk approach is more computational intensive (Table 2) especially when scaling up the sequence length.

**Questions For Authors:**

I don't have major questions.

**Relation To Broader Scientific Literature:**

Being able to edit long-form, diverse formats knowledge in LLM is still under-explored. There are a few works (e.g. UnKE) for long-form knowledge edits, but the paper shows clear improvements.

**Theoretical Claims:**

I carefully read the proofs in the main paper. I briefly scanned through the proof in appendix.

---

> ### Author Rebuttal · Authors · 2025-03-28
>
> Dear Reviewer hVmV:
>
> Thank you for your kind words and positive feedback regarding the novelty, presentation, and effectiveness of our work! Your approval is the great encouragement for us and motivates us to continue advancing our research.
>
> Below, we meticulously provide responses to each of your comments and outline the modifications based on your suggestions.
>
> ## *W1: "The proposed chunk by chunk approach is more computational intensive especially when scaling up the sequence length."*
>
> Thank you for raising this important concern. We acknowledge that our method indeed faces computational intensity issues, particularly when scaling up sequence length, and we highlighted this explicitly in Section 5.4 (Observation 6) of our paper. We have actively been exploring optimization strategies to address this challenge:
> 1. **Early stopping strategy:** During gradient descent optimization of Equation (9), we halt training once loss thresholds are met, allowing shorter optimization epochs without sacrificing performance. Due to shorter chunk lengths, thresholds are quickly reached, significantly accelerating the process.
> 2. **Adaptive chunk length selection:** Chunks vary in difficulty; simpler chunks can be longer, while challenging chunks can be shorter. Combined with loss thresholds, this significantly reduces the gradient descent epochs.
> Additionally, since our initial submission, we have conducted preliminary experiments in the first direction and observed improvements in editing speed. A brief overview of these initial experimental results is provided in the following table:
>
> |||||
> |:-:|:-:|:-:|:-:|
> |Method|UnKEBench|Counterfact|MQUAKE|
> |MEMIT|16.37|17.05|16.92|
> |AnyEdit|21.14|20.43|21.28|
> |AnyEdit(accelerate)|18.89|8.22|19.03|
> |||||
>
> As shown, our optimization approach significantly improves editing speed over the original chunk-by-chunk method, and now **matches the performance of the single-token editing baseline**. We will continue working on further improving the chunk-by-chunk editing efficiency.
>
> We hope this response addresses your concerns.
>
> ## *Suggestion1: "I recommend the authors to also include DEM in the experiments for more comprehensive comparison."*
>
> Thank you for highlighting this important method! Your insightful suggestion has prompted us to recognize the value of including DEM for a more comprehensive evaluation. In response:
>
> 1. We have provided a detailed description of DEM in the Experimental Setup section of the revised manuscript.
> 2. We conducted additional experiments using DEM and presented the corresponding results and analysis in Section 5.2.
>
> Note: Since DEM's code is currently not publicly available, we implemented the method ourselves according to our interpretation from the original paper. Below is a quick summary of our experimental results and analysis:
>
> |||||||||||||
> |:-:|:-:|:-:|:-:|:-:|:-:|:-:|:-:|:-:|:-:|:-:|:-:|
> |LLM|Method||**UnKEBench**||||**Counterfact**|||**MQUAKE**||
> |||**Ori.**|**Ori.**|**Paraph.**|**Paraph.**|**Ori.**|**Ori.**|**Paraph.**|**Paraph.**|**Ori.**|**Ori.**|
> |||BertScore|Rouge-L|BertScore|Rouge-L|BertScore|Rouge-L|BertScore|Rouge-L|BertScore|Rouge-L|
> |Llama3-8B-It|DEM|77.09±0.32|31.27±0.48|75.18±0.29|29.41±0.57|77.01±0.30|32.87±0.47|48.12±0.36|16.32±0.54|75.96±0.32|23.10±0.57|
> |Llama3-8B-It|AnyEdit|**97.76±0.11**|**92.96±0.24**|**96.60±0.19**|**95.60±0.35**|**97.76±0.14**|**95.87±0.23**|**62.63±0.44**|**46.51±0.59**|**96.33±0.21**|**94.32±0.23**|
> |Qwen2.5-7B-It|DEM|78.92±0.28|38.71±0.50|77.01±0.25|29.32±0.48|78.10±0.27|39.50±0.46|56.32±0.35|26.09±0.53|74.18±0.30|35.25±0.52|
> |Qwen2.5-7B-It|AnyEdit|**98.05±0.16**|**94.89±0.29**|**93.56±0.15**|**79.98±0.28**|**98.08±0.15**|**95.09±0.19**|**65.40±0.38**|**43.49±0.47**|**98.14±0.13**|**96.39±0.18**|
> ||||||||||||
>
> As the table demonstrates, our AnyEdit method consistently outperforms DEM. Although DEM dynamically selects layers due to common sense knowledge residing across different layers, it still operates under the limitation of single-token editing and thus struggles with long-form knowledge editing.
>
> We greatly appreciate your valuable suggestions and believe these modifications significantly strengthen our paper.
>
> ---
>
> Once again, we deeply appreciate your thoughtful and encouraging feedback. Your suggestions have not only enhanced the current work but have also inspired us to continue exploring research in the area of model editing. We are excited to keep moving forward and contributing to the community!
>
> Best,
>
> Authors of Submission9475

---

> > ### Comment · Reviewer_hVmV · 2025-04-02
> >
> > The authors' rebuttal addresses my concerns. I will keep my score.

---

> > > ### Author Response · Authors · 2025-04-03
> > >
> > > Dear Reviewer hVmV,
> > >
> > > Thank you very much for your feedback and for keeping the score. We truly appreciate your support and encouragement. Your positive evaluation of our work means a great deal to us, and we are grateful for your time and thoughtful review.
> > >
> > > We look forward to any further suggestions you may have in the future.
> > >
> > > Best regards,
> > >
> > > Authors of the Paper 9475

---

### Decision · Program_Chairs · 2025-05-01

**Decision:**

Accept (poster)

**Comment:**

This work proposes AnyEdit, a method that addresses the gap in long-form, diverse-format knowledge editing for LLMs. Reviewers particularly appreciated the clear theoretical grounding via the Chain Rule of Mutual Information and the strong empirical evaluations across multiple benchmarks, including the introduced EditEverything dataset.

During the rebuttal phase, the authors effectively addressed the reviewers’ initial concerns through detailed responses, providing additional analyses and comparisons with previously unaddressed baselines. The reviewers unanimously gave the paper positive scores and emphasized the practical importance of extending from triplet-based to free-form knowledge editing.

However, the paper still has the following issues, which the authors should incorporate into the revision version, especially considering the discussion during the rebuttal:
1. Computational Efficiency, Reviewer hVmV pointed out that the proposed chunk-by-chunk approach remains computationally intensive despite optimization efforts, potentially limiting its practicality for very long sequences. Although the authors suggested some possible tricks, this remains a major drawback of the method. The authors need to discuss this issue in more detail in the revised version.
2. Scope of Evaluation, Reviewer H3vg noted the limitation of evaluating the method only on sequences up to 200 tokens, which does not fully support the claim of arbitrary-length editing. The authors did not respond to this point in the rebuttal.
3. Broader Impact Analysis, The reviewers raised valid concerns about the impact of extensive edits on unrelated knowledge. A more comprehensive assessment in this area would further strengthen the work.

In summary, the committee finds this paper clearly worthy of acceptance due to its novel contributions and rigorous evaluations. However, it recommends acceptance without an oral presentation, encouraging the authors to further address computational efficiency and broader validation in future work.